# Reduced contribution of sulfur to the mass extinction associated with the Chicxulub impact event

Katerina Rodiouchkina [1,2,3] ✉, Steven Goderis [2], Cem Berk Senel [2,4], Pim Kaskes [2,5], Özgür Karatekin[4], Michael Ernst Böttcher [6,7,8], Ilia Rodushkin[3,9], Johan Vellekoop [10,11], Philippe Claeys [2] & Frank Vanhaecke [1]

The Chicxulub asteroid impact event at the Cretaceous-Paleogene (K-Pg) boundary ~66 Myr ago is widely considered responsible for the mass extinction event leading to the demise of the non-avian dinosaurs. Short-term cooling due to massive release of climate-active agents is hypothesized to have been crucial, with S-bearing gases originating from the target rock vaporization considered an important driving force. Yet, the magnitude of the S release remains poorly constrained. Here we empirically estimate the amount of impact-released S relying on the concentration of S and its isotopic composition within the impact structure and a set of terrestrial K-Pg boundary ejecta sites. The average value of $67 \pm 39$ Gt obtained is ~5-fold lower than previous numerical estimates. The lower mass of S-released may indicate a less prominent role for S emission leading to a milder impact winter with key implications for species survival during the first years following the impact.

Approximately 66 million years ago, a 10–15 km sized carbonaceous chondritic asteroid[1,2] collided with Earth, on the northern Yucatán Peninsula in Mexico, forming the ~200 km in diameter Chicxulub crater[3]. This event likely triggered the mass extinction of approximately 75% of all species[4], including the non-avian dinosaurs, and led to the near-global loss of vegetation[5,6] at the Cretaceous-Paleogene (K-Pg) boundary[5–11]. The mass extinction is hypothesized to be due to a rapid change of climatic conditions, resulting from the massive release of dust and climate-active gases, including water vapor, $CO_2$, $CH_4$, and sulfur (S)-bearing gases, by shock-vaporization of the seawater and the carbonate- and evaporite-rich sedimentary cover of the Yucatán target rock[5,9,12,13]. In addition, a massive release of fine-grained ejecta reduced the amount of solar radiation reaching the Earth's surface, leading to a global impact winter state[13]. These fine-grained ejecta consist of silicate dust originating from the pulverization of the deep Yucatán granitic basement following the impact[5,13,14], sulfate aerosols formed from the vaporized evaporites and seawater[15], and soot from buried hydrocarbons[16,17] and possible impact-induced wildfires[16,18–20]. The thus-induced impact winter triggered extremely cold conditions and a blockage of photosynthesis that affected Earth at the scale of years to decades[13,21–24]. Current models consider sulfate aerosols to be a crucial contribution to this global perturbation as they induce a net cooling effect due to their strong backscattering of solar radiation[25,26]. The impact winter is followed by long-term gradual global warming for tens of thousands of years steered by $CO_2$ emitted from the dissociation of the carbonate target[5,12,21–24,27–30]. Short-term global cooling is often regarded as the most critical step in the global extinction event, although the

[1]Atomic and Mass Spectrometry—A&MS research unit, Department of Chemistry, Ghent University, Ghent, Belgium. [2]Archaeology, Environmental Changes & Geo-Chemistry, Vrije Universiteit Brussel, Brussels, Belgium. [3]Division of Geosciences, Luleå University of Technology, Luleå, Sweden. [4]Reference Systems and Planetary Department, Royal Observatory of Belgium, Uccle, Belgium. [5]Laboratoire G-Time, Université Libre de Bruxelles, Brussels, Belgium. [6]Geo-chemistry & Isotope Biogeochemistry, Leibniz-Institute for Baltic Sea Research (IOW), Warnemünde, Germany. [7]Marine Geochemistry, Greifswald University, Greifswald, Germany. [8]Interdisciplinary Faculty, University of Rostock, Rostock, Germany. [9]ALS Scandinavia AB, ALS Laboratory Group, Luleå, Sweden. [10]Division of Geology, KU Leuven, Leuven, Belgium. [11]Institute for Natural Sciences, Brussels, Belgium. ✉e-mail: Katerina.Rodiouchkina@gmail.com

volume and relative role of impact-released sulfur remain poorly constrained.

The amount of sulfate aerosols injected within the atmospheric column and the residence time of the corresponding particles determine the severity of the sulfur-induced global cooling[30]. Residence time in the atmosphere depends on the altitude to which the impact-vaporized S is injected into the atmosphere[12] and the type of S-species injected[26]. Impact-vaporized S-species injected into the oxidizing Paleogene atmosphere[31] react with $O_2$, $OH^-$, and/or $H_2O$ to form sulfate aerosol ($H_2SO_4$), usually through the following series of simplified reactions $SO_2 + OH \rightarrow HSO_3^- + O_2 \rightarrow SO_3 + H_2O \rightarrow H_2SO_4$[26]. Photodissociation reactions also occur due to the absorption of visible light and/or UV-radiation following $H_2SO_4 + h\nu \rightarrow SO_3 + H_2O$[26] and $SO_2 + h\nu \rightarrow SO + O$[32]. With time, the resulting sulfate aerosols settle down onto the Earth's surface via wet and/or dry deposition mechanisms. Reduced S-species have a longer residence time than oxidized species, and the injection height of the particles influences the reaction rates as the atmospheric conditions at higher altitudes are less oxidizing, drier, and less shielded from sunlight (with residence times of days to weeks for the troposphere and years to decades for the stratosphere)[26,30,33]. Numerical modeling of these physical and chemical processes in the climate system documents the environmental changes triggered by the impact. The accuracy of paleoclimate model studies focusing on the atmospheric influence of the impact-released S[13,23,24] highly depends on adequate estimations of the total amount of impact-volatilized S.

Current estimates of impact-released S rely on numerical methods that simulate the impact, using assumed target-rock and projectile parameters based on field observations and small-scale laboratory experiments to constrain the amount released into the atmosphere. The earliest estimates of total volatilized S by Sigurdsson et al. in 1992 (4300 Gt S)[15], Brett in 1992 (200 Gt S)[34], and Chen et al. in 1994 (90 Gt S)[35] (see Table 1 for more detailed information on the model assumptions and outcome) are now considered unreliable as they are based on incomplete geological records, inaccurate assumptions about impact parameters, or invalid experiments involving devolatilization of calcium sulfate. The subsequent estimates obtained by Pope et al. in 1994 (35–210 Gt S)[22] and Ivanov et al. in 1996 (6–106 Gt S)[36] (Table 1) both used the same impact model, but these studies were performed using outdated constraints for the composition, size, and equations of state (EOS) for both the impact site and projectile, while they also did not consider the obliqueness of the impact (impact angle to the surface normal)[21,36]. Rudimentary atmospheric modeling based on the estimates from Pope et al. 1994[22] and Ivanov et al. 1996[36] was carried out by Pope et al. in 1997[37] (Table 1).

Pierazzo et al. 1998[21] addressed some of the shortcomings of the former numerical models for estimating the amount of impact-released S (152–253 Gt S) by considering the size, velocity, and porosity of the projectile, as well as target rock stratigraphy and updated EOS for the anhydrite in the target (Table 1). As laboratory experiments had in the meantime shown that 30–60% of vaporized S from anhydrite/gypsum would be trapped as condensate through recombination with CaO in the cooler parts of the impact plume[36], Pierazzo et al. 1998[21] roughly assumed that 50% of all the vaporized S would be lost to recombination effects before being injected into the atmosphere (76–127 Gt S). In 2003, Pierazzo et al.[23] simulated the resulting climate changes by applying the estimates of released S as obtained in Pierazzo et al. 1998[21] and ignoring other climate-active gases and particles (Table 1). More than a decade later, Brugger et al. 2017[28] performed further climate modeling closely following the climate model of Pierazzo et al. 2003[23] (Table 1).

In 2017, Artemieva et al.[12] applied more advanced impact simulation models to estimate the total load of impact-volatilized S into the atmosphere (Table 1), adding recent constraints on the shock pressure, the composition of the target rock, as well as asteroid impact angle.

This latest simulated estimate provided an estimated amount of $325 \pm 130$ Gt of released S, which is higher than any previous estimates, except for Sigurdsson et al. 1992[15]. To date, the simulations of the Chicxulub impact and amount of impact-released S by Artemieva et al. 2017[12] are considered the most reliable. Tabor et al. 2020[24] and Senel et al. 2023[13] used the numerical estimation of 325 Gt S from Artemieva et al. 2017[12] in two short-term paleoclimate studies in which the individual and combined effects of S, soot, and silicate or iron dust released following the Chicxulub impact event were modeled (Table 1).

The reliability of numerical estimates is heavily reliant on assumptions about the Chicxulub impact shock pressure and the percentage of S that this shock pressure released from the target rock, which in part explains the large range of published estimates (6 to 4300 Gt S, Table 1). Previously used numerical estimations of impact-released S also rely heavily on the assumption of the proportion and distribution of evaporite within the target (Table 1). Considering the size of the impact structure (~200 km) and the limited volume of impactite recovered during several drilling projects[29], a rigorous estimation of the proportion of evaporites present in the target rock lithology remains challenging. Currently, this estimation mainly relies on the stratigraphy (~27% anhydrite with the remaining intervals dominated by limestones and dolomites) of the ICDP (International Continental Scientific Drilling Program) Yax-1 (Yaxcopoil-1) drilling in 2001–2002 that penetrated the undisturbed upper half of a ~3 km sequence of target Cretaceous sediments located beneath the outer annular trough of the Chicxulub impact structure[38–40]. The lower half of the Yucatàn sedimentary target is estimated to be more evaporite-rich (22–63% anhydrite) based on geophysical logs and well cuttings of the PEMEX (Petróleos Mexicanos) boreholes sampled in the 1950s and 1960s south/east of the crater[21,41] and stratigraphy of the 1994–1995 UNAM (Universidad Nacional Autónoma de México) drill cores within the ejecta blanket south of the Chicxulub impact structure[42,43]. However, these drill cores outside the crater rim may be too shallow to be representative of the deepest part of the sedimentary target rock. In contrast, the 2016 offshore drill core M0077A recovered from the northern part of the Chicxulub peak ring by IODP (International Ocean Discovery Program)-ICDP Expedition 364 is largely devoid of gypsum and anhydrite, and sulfur-bearing phases are limited to pyrite, chalcopyrite, and minor accessory minerals[29,44,45]. The potential heterogeneous distribution of evaporite within the Yucatán target constitutes a considerable source of uncertainty during numerical estimation, which makes this approach to document S release unreliable.

For a more reliable estimate of the S released, the empirical estimates applied in the current study, instead of focusing on the impact event itself, rely on the aftermath of the impact-vaporized S injected into the atmosphere, mainly the wet and dry deposition of the globally distributed sulfate aerosols back to Earth's surface. If massive amounts of impact-sulfate aerosols are deposited globally, a positive S concentration anomaly should be observed at and after the K-Pg boundary clays in the sedimentary profiles of known K-Pg boundary sites. The amount of S in these K-Pg boundary sediment profiles related to the impact can be quantified using an isotope dilution approach if the S isotopic fingerprint ($\delta^{34}S$) of the target rock evaporite is constrained with sufficient precision and if it is sufficiently distinct compared to the $\delta^{34}S$ of the natural bedrock of the K-Pg boundary site. These S anomaly profiles allow one to empirically estimate the total amount of impact-released S using mass balance calculations. In contrast with previous numerical estimations[12,15,21,22,34–36], assumptions about the impact angle, projectile size, velocity, shock pressures, target rock stratigraphy and porosity, and interactions within the impact-related gas plume are not required. Instead, calculations rely on the S isotope ratio of the target rock and the preservation of the deposition record.

To date, only a few studies have examined S isotopic compositions across the K-Pg boundary, where S isotope ratios in combination

**Table 1 | Comparison of previously published numerical estimates for S release and associated climatic effects**

| | Estimates | Calculation of estimate | Assumptions | Influence on climate |
|---|---|---|---|---|
| Sigurdsson et al. 1992[15] | 4300 Gt S (180 km-diameter impact crater)[a] | Based on the size of the Chicxulub impact crater | – 2 km-thick sedimentary layer with 50% evaporites in the target<br>– Complete degassing at shock pressures ≥ 40 GPa | N/A |
| Brett 1992[34] | 200 Gt S (150 km to a 180 km-diameter crater)[b] | Based on scaling-up simulations from Roddy et al. 1987[19] of a 10 km-diameter projectile traveling at 20 km/s | – 500 m anhydrite layer in the target<br>– Anhydrite decomposition at temperatures of ≥ 1800 K | N/A |
| Chen et al. 1994[35] | 90 Gt S (asteroidal impactor)[b] | Asteroidal impactor pressure decay modeling | – 300 m $CaSO_4$ bed with 50% anhydrite<br>– Portions of degassing: 2% of solid, 100% of melt, and 100% vapor reaction phases of anhydrite/gypsum at a shock pressure of 42 GPa | Global cooling by 5–9 or 10–19 °C based on extrapolation of global cooling from volcanic S release power function |
| Pope et al. 1994[22] | 35–210 Gt S (180 km-diameter crater, 10 km-diameter impactor traveling at 20 km/s)[c] | Two-dimensional (2-D) hydrocode vertical impact model of a two-layer target | – 500–1500 m-thick anhydrite layer<br>– Complete degassing of anhydrite at shock pressures >100 GPa | Solar transmission decreases to 20–10% for 8–13 yr post-impact (radiative transfer model designed for studies of planetary atmospheres) |
| Ivanov et al. 1996[36] | 6–106 Gt S (180 km-diameter crater, 10 km-diameter impactor traveling at 20 km/s, lowest value from vaporized anhydrite outside projectile footprint)[c] | The same 2-D hydrocode vertical impact model as in Pope et al. 1994[22] | – 23–60% anhydrite<br>– Complete degassing of anhydrite at shock pressures >100 GPa | Solar transmission decreases to 20–10% during the first year, followed by 50% during the following 8–13 yr[37] |
| Pierazzo et al. 1998[21] | – 152–253 Gt S (~ 100-km-diameter transient cavity, 15 km-diameter asteroid traveling at 20 km/s)<br>– 76–127 Gt S (assuming 50% of the vaporized S is lost globally to recombination effects, based on laboratory experiments[36])[d] | 2-D hydrocode vertical impact model with new EOS for anhydrite | – 30–50% evaporites in target<br>– Complete degassing of anhydrite at shock pressures >100 GPa | – Global cooling by 2–8 °C with prolonged effects lasting up to a decade, with saturation effect at higher estimates (> 30 Gt S)[23]<br>– Global cooling of 27–30 °C, reaching minimum temperatures 3 yr post-impact with full recovery to 30 yrs[28] |
| Artemieva et al. 2017[12] | 325 ± 130 Gt S (90–100 km-diameter transient cavity, 10 km-diameter impactor traveling at 18 km/s)[e] | Three-dimensional (3-D) hydrocode impact model with 60° ±10° obliqueness | – The upper half of the sedimentary sequence was 25% anhydrite and the lower half 60%<br>– Incipient decomposition of anhydrite at 30 GPa and full decomposition at 120 GPa (linear interpolation in-between) | – Global cooling by 14 °C during the first 4 yrs after impact[24] or by 24 °C during the first 1.5 yrs after impact[13]<br>– Photosynthetic shut down mainly due to dust, not S[13] |

[a] Now considered a gross overestimation, due to severe underestimation of the required shock pressures for complete degassing of anhydrite.
[b] Based on currently assumed incomplete geological records, inaccurate impact parameters, and invalid experiments involving the decomposition of anhydrite.
[c] Relying on outdated constraints for the composition, size, and EOS (equations of state) for both the impact site and projectile. The Obliqueness of the impact (impact angle to the surface normal) was not considered.
[d] Obliqueness of the impact was not considered and assumes partial vaporization of anhydrite <100 GPa.
[e] Result of improved computational power, possibility to model obliqueness, and longer simulations (15–30 s post-impact, instead of a couple of s). However, no EOS for the anhydrite was included, and no consideration of recombination effects that could influence the total amount of globally spread impact-released S. Widely considered the most reliable estimate to date.

with S concentrations have been mostly used to investigate post-impact changes to the biogeochemical S-cycle. This includes shifting sulfate levels in seawater[46,47], duration of oxic/anoxic conditions based in part on observed large negative $\delta^{34}$S shifts due to bloom of sulfate-reducing microbes[48–54], and input from impact-generated atmospheric sulfur[30,49,52,55,56]. None of these studies used the empirical S data from a pristine K-Pg boundary depositional site to estimate the total amount of impact-released S.

The current study provides an empirical estimate of the amount of impact-vaporized S using an isotope dilution approach based on a set of K-Pg boundary deposition sites to study the role of S in triggering the post-impact winter. The S fingerprint of the target rock is constrained by investigating the S content and isotopic composition in drill cores located within the Chicxulub impact structure and the proximal ejecta blanket. Sulfur concentrations and $\delta^{34}$S values in K-Pg boundary sections, varying from terrestrial to deep marine, and proximal to distal relative to the impact-site, are determined to yield full geochemical profiles.

## Results and discussion
### Sulfur concentration and isotopic composition in the target
For the applied isotope dilution approach at the global scale, the first step is to constrain the S isotopic fingerprint ($\delta^{34}$S) of the target rock evaporite. The concentration of S and its isotopic composition were determined in selected lithological units in Cretaceous sediments of five onshore drill cores: PEMEX Y6 (Yucatán 6, located inside the annular crater moat surrounding the Chicxulub peak ring area ~ 50 km southwest from the center of the impact structure)[57,58], UNAM-5 (located outside the crater in the proximal ejecta blanket ~ 105 km south from the center of the impact structure), UNAM-6 (located outside the crater in the proximal ejecta blanket ~ 151 km southeast of the center of the impact structure), UNAM-7 (located outside the crater in the proximal ejecta blanket ~ 126 km southeast of the center of the impact structure)[42], and ICDP Yax-1 (Yaxcopoil-1, located in the outer part of an annular trough ~ 60 km south of the center of the impact structure)[38], as well as the offshore drill core IODP-ICDP Expedition 364 M0077A (located on the topographic peak ring ~ 46 km northwest of the center of the impact structure)[44]. Figure 1A shows the locations of these drill cores within the Yucatán peninsula. Measured bulk $\delta^{34}$S values and S concentrations throughout the different lithological units of the M0077A drill core are shown in Fig. 1B (Supplementary Fig. S1 in the Supplementary information, SI, focuses on the K-Pg boundary claystone interval only). In addition, Fig. 1 represents the range in $\delta^{34}$S values and S concentrations for the five onshore drill cores.

Anhydrite within unshocked Cretaceous sediments at the impact site (Yax-1) and in the proximal ejecta blanket (UNAM-5, 6, and 7) vary between 6 and 23 wt% S and $\delta^{34}$S values range between 18.0 and 19.5‰. Suevite and impact melt breccia intervals within drill core Y6 contain 0.5 to 2 wt% S with $\delta^{34}$S between 17.1 and 17.9‰ (Fig. 1B, Supplementary Fig. S2 and Supplementary Table S1 in the SI). All onshore drill cores display $\delta^{34}$S values (17.1–19.5‰) that agree with the previously determined average $\delta^{34}$S value of 18.3‰ for the Yax-1 and Y6 drill cores (ranging between 18.0 to 19.8‰)[59] and the seawater sulfate $\delta^{34}$S values ranging between 17 to 19‰ at the end of the Cretaceous[46,60]. The high S concentrations observed for Yax-1 and UNAM-6 (23 and 20 wt% S, respectively) indicate pure anhydrite and gypsum (19–24 wt% S), while the S content in the other drill core samples (0.5–10 wt% S) represents a mixed composition with a large admixture of evaporite as even the lower S concentrations found in the suevite and impact melt breccia rock sections of the Y6 core have similar $\delta^{34}$S values (Supplementary Fig. S2 in the SI). The total reduced inorganic S (TRIS) content is measured for the suevite and impact melt breccia sections in the Y6 drill core (24 and 52 $\mu g\,g^{-1}$, respectively, Supplementary Table S3 in the SI). This corresponds to TRIS fractions of 0.1 and 1% of the bulk S

concentration, respectively, indicating that these sections mostly contain oxidized S-species such as anhydrite and gypsum.

Detailed bulk S concentration and $\delta^{34}$S profiles are provided for M0077A as evaporites had previously been observed to be largely absent from this drill core[29], indicating effective degassing of anhydrite within the peak ring of the impact crater or post-impact dissolution following enhanced fluid flow in the hydrothermal system of the impact basin. These profiles span different lithological units[45], including the post-impact Paleogene sediments, suevite (impact-melt bearing polymict breccia), upper impact melt rock (UIM), lower impact melt rock-bearing rock (LIMB), and intermediate intervals below the UIM, including granitoid, dolerite, and dacite basement material to constrain the various contributing sources of S and determine the effects of post-impact processes overprinting the primary S isotopic signatures (Fig. 1B and Supplementary Table S1 in the SI, detailed discussion found in SI). The TRIS fraction in selected samples throughout the M0077A drill core (140–7000 $\mu g\,g^{-1}$) corresponds to 22–86% of the bulk S concentration. The lowest TRIS fractions (22–33%) are found in the graded suevite, a metamorphic clast, and lower LIMB units (Supplementary Fig. S5 and Supplementary Table S3 in the SI, detailed discussion found in SI), consistent with higher bulk $\delta^{34}$S values (0.7–8.5‰), indicating that traces of evaporite may be preserved within the M0077A drill core. Among them, the graded suevite section has the highest bulk $\delta^{34}$S value and the largest shift (14.0‰) between the bulk $\delta^{34}$S and the sulfide-specific $\delta^{34}$S (− 5.5‰, Supplementary Table S3 in the SI). As the bulk S concentrations (500–1500 $\mu g\,g^{-1}$) and $\delta^{34}$S values are lower in these lithological units compared to the five onshore drill cores (Yax-1, Y6, and UNAM-5-7), the signals measured for the target bedrock are likely influenced by later post-impact processes. For instance, in contrast to the graded suevite section, the bedded suevite section right above has a significantly lower bulk $\delta^{34}$S (− 7.4 to − 5.3 ‰), a higher TRIS fraction (85%), and a small shift (1.9‰) between the bulk $\delta^{34}$S and the sulfide-specific $\delta^{34}$S (− 9.2‰) (Supplementary Tables S1 and S3 in the SI). Similarly low sulfide $\delta^{34}$S values have previously been observed in the post-impact sediment sections in the M0077A drill core by Schaefer et al. 2020[53] (Fig. 1B) and suggested by Kring et al. 2020[61], 2021[54] to result from late-stage microbial reduction of S in impact-generated hydrothermal systems. Following the effective overprinting of various drill core intervals, the bulk $\delta^{34}$S value of the target rock is best determined based on the $\delta^{34}$S values measured for the evaporite-containing lithological units in five different drill cores located within and around the Chicxulub impact structure (Yax-1, Y6, and UNAM-5-7), which leads to an average target rock $\delta^{34}$S value of 18.5 ± 1.4‰ (2 SD).

### Sulfur concentration and isotopic profiles for K-Pg boundary sites
The second step in the global scale isotope dilution approach is to find preserved positive S concentration anomalies within the sediment profiles of K-Pg boundary deposition sites indicating airfall of the impact-vaporized S from the target rock. As the average impact target rock $\delta^{34}$S value is ~18.5‰ (±1.4‰, 2 SD), positive $\delta^{34}$S values are expected to coincide with the elevated S concentrations found within sediment profiles unless the natural bedrock of the site has similar or higher $\delta^{34}$S values as/than the target rock. The locations of all the K-Pg boundary deposition sites included in this study are presented in Fig. 2A. Figure 2B–I shows the bulk S concentration and $\delta^{34}$S profiles. Full profiles across the K-Pg boundary are measured for four of these K-Pg boundary sites (Fig. 2B–E). The localities selected for full profiles include the distal mid-shelf marine site at Stevns Klint (Denmark, neritic with estimated water depths at K-Pg boundary times of 100–150 m[62], Fig. 2B), a distal deep marine site at Caravaca (Spain, bathyal with estimated water depths of 500–1000 m[62], Fig. 2C), a proximal mid-shelf marine K-Pg boundary section at Brazos River (Texas, USA, neritic with estimated water depths of 75–100 m[10], which

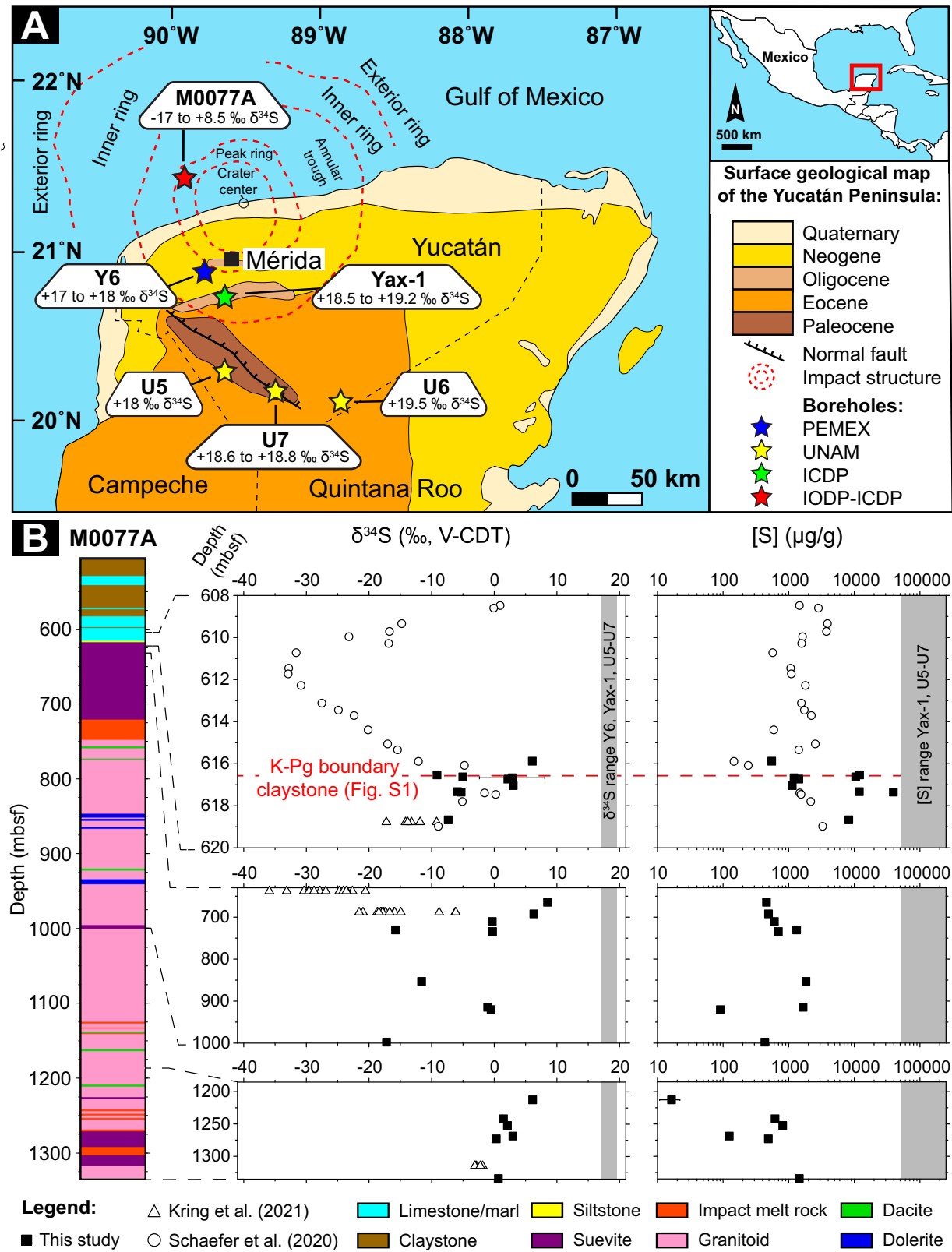

is a well-established tsunami deposit (tsunamite)[63–66], Fig. 2D), and an intermediate terrestrial site at Tanis (Hell Creek Formation, North Dakota, USA[67], Fig. 2E) located approximately 6400; 5500; 900; and 3000 km from the Chicxulub impact structure, respectively. For these sites, airfall deposition following the impact event is additionally assessed by measuring siderophile element concentrations, including Ir, Re, Cr, Co, and Ni, and ratios between these element concentrations

in impact-related sediment profiles (Fig. 3, Supplementary Figs. S6–8 and Supplementary Table S4). The presence of a meteoritic contribution is confirmed when the concentration of Ir and the other siderophile elements are highly elevated compared to background (i.e., continental crustal) values, and the ratios between these elements are roughly similar to chondritic ratios in impact-related lithologies (melt rocks, impact ejecta material, etc.)[68]. For comparison, the bulk S

**Fig. 1 | Summarized S data for drill cores located within and close to the Chicxulub peak ring area. A** Simplified geological surface map of the Yucatán Peninsula in Mexico (modified from Rebolledo-Vieyra and Urrutia Fucugauchi 2004[120], Kring 2005[121], and Kaskes et al. 2021[45]; used with permission of The Geological Society of America) with the subsurface features of the Chicxulub impact structure marked. On the map, drill core locations are highlighted using stars of different colors: Y6 (Yucatán 6) from PEMEX (Petróleos Mexicanos); U5 (UNAM-5), U6 (UNAM-6), and U7 (UNAM-7) from UNAM (Universidad Nacional Autónoma de México); Yax-1 (Yaxcopoil-1) from ICDP (International Continental Scientific Drilling Program); and M0077A from IODP-ICDP (International Ocean Discovery Program- International Continental Scientific Drilling Program) Expedition 364. Next to each star, the corresponding measured bulk $\delta^{34}$S range is presented.

**B** Simplified lithological column for the IODP-ICDP Expedition 364 M0077A drill core with corresponding measured bulk $\delta^{34}$S values and S concentrations (on a logarithmic scale). The equivalent of the K-Pg (Cretaceous-Paleogene) boundary claystone, characterized by an Ir anomaly based on Goderis et al. 2021[117], is shown here by a red dashed line. A higher resolution visualization of the interval surrounding the K-Pg boundary claystone is displayed in Supplementary Fig. S1. For comparison, previously published pyrite-related S concentrations and $\delta^{34}$S values from two studies of the M0077A drill core are included, Schaefer et al. 2020[53] (circular markers) and Kring et al. 2021[54] (triangular markers). The range of measured bulk $\delta^{34}$S values and S concentrations for the five other drill core samples (Y6, Yax-1, U 5–7, and Yax-1, U 5–7, respectively) obtained in the present study are highlighted by a gray area in the graphs.

concentration and the $\delta^{34}$S value were additionally investigated for other K-Pg boundary sites but using only a single sample located exactly within the K-Pg boundary claystone (Supplementary Table S2 in the SI). These sites included a proximal bathyal marine site at Beloc (Haiti, $150\,\mu g\,g^{-1}$ and 17 ‰), several intermediate terrestrial sites at Long Canyon (Raton Basin, Colorado, USA, $1500\,\mu g\,g^{-1}$ and 4.0 ‰), Dogie Creek (Powder River Basin, Wyoming, USA, $3300\,\mu g\,g^{-1}$ and − 0.89 ‰), Brownie Butte (Hell Creek area, Montana, USA, $4200\,\mu g\,g^{-1}$ and − 3.2 ‰), and Seven Blackfoot Creek (Hell Creek area, Montana, USA, $500\,\mu g\,g^{-1}$ and − 2.0 ‰), as well as several distal bathyal and outer-neritic/upper-bathyal marine sites at Frontale (Italy, $90\,\mu g\,g^{-1}$ and 16 ‰), Fonte d'Olio (Italy, $170\,\mu g\,g^{-1}$ and 16 ‰), Siliana (Tunisia, $3600\,\mu g\,g^{-1}$ and 13 ‰), and Elles (Tunisia, $63,000\,\mu g\,g^{-1}$ and 18 ‰) located approximately 500, 2250, 3000, 3100, 3100, 6300, 6300, 6700, and 6700 km from the Chicxulub impact structure[68], respectively (Fig. 2A).

A positive peak in S concentration coinciding with positive $\delta^{34}$S values, as expected for atmospherically deposited Yucatán target anhydrite, is observed in the boundary claystone and coal interval of the terrestrial Tanis K-Pg site ($600–8000\,\mu g\,g^{-1}$ and − 6 ‰ to 5 ‰, respectively, Fig. 2E; Supplementary Table S2 in the SI). This positive S anomaly coincides with increases in the Co, Cr, Ni, Ir, and Re concentrations in the post-impact claystone sediment (Fig. 3 and Supplementary Table S4). The Ni/Cr ratios (Fig. 3 and Supplementary Table S4) around the S increase for the Tanis site range between 1.6 and 3.4, which is distinct from the Ni/Cr values of the Earth crust around ~ 0.5[68,69]. This range indicates meteoritic input as these values are more similar to the Ni/Cr values of 1.1–3.4 previously observed in smectites from a K-Pg boundary site at Beloc, Haiti[70]; the slope of 4.3 ± 1.3 (Ni/Cr range between 0.09–7.42) for the regression line of Ni and Cr values for 48 different K-Pg boundary ejecta layers from all around the globe[68]; and the mean Ni/Cr for carbonaceous chondrites of CO-type (3.96 ± 0.18, 2 SD), CM-type (4.01 ± 0.60, 2 SD), and CR-type (3.72 ± 0.78, 2 SD)[71]; which have been suggested to match the Chicxulub impactor[1]. The previously published median grain sizes for the Tanis K-Pg sediment profile[13] lack data in the Paleocene coal interval coinciding with the positive S anomaly.

The median grain size immediately below the Paleocene coal interval coinciding with the positive S anomaly (in the claystone interval, right above the K-Pg boundary line in Fig. 3) is significantly finer (2.88–3.83 μm) and unimodal compared to the event deposit siltstone right below (10.63–11.27 μm) and the Paleocene siltstone above (12.75–29.86 μm). This finer grain size has previously been reported to result from atmospheric settling[13], although further sedimentological studies and grain-size measurements in the coal interval should be carried out. Combining the Ir and Re information data with the Co, Cr, and Ni concentrations, Ni/Cr ratios, and grain-size profile, the increase in S concentrations and $\delta^{34}$S values observed at the Tanis site likely indicates airfall of finely mixed impactor and target rock material directly on top of the Tanis silty event deposit[67].

The positive S offset observed in the S profile for the Tanis site is similar to the previously documented S-profiles for other terrestrial

K-Pg sites. These sites include Dogie Creek (~ $500\,\mu g\,g^{-1}$ to ~ $26,000\,\mu g\,g^{-1}$ and − 5 ‰ to 1 ‰, Fig. 2F, Ir-anomaly previously published[68]) and Brownie Butte (~ $2000\,\mu g\,g^{-1}$ to ~ $22,000\,\mu g\,g^{-1}$ and − 3 ‰ to 6 ‰, Fig. 2G, Ir-anomaly previously published[68]) in Maruoka et al. 2002[52] and Knudsen's Coulee (Canada at ~ 4300 km, ~ $300\,\mu g\,g^{-1}$ to ~ $9000\,\mu g\,g^{-1}$ and 2 ‰ to 17 ‰, Fig. 2H, Ir-anomaly previously published[72,73]) and Knudsen's Farm (Canada at ~ 4300 km, ~ 2.5 km from Knudsen's Coulee site, ~ $500\,\mu g\,g^{-1}$ to ~ $8000\,\mu g\,g^{-1}$ and 2 ‰ to 9 ‰, Fig. 2I, Ir-anomaly previously published[72,73]) in Cousineau 2013[56]. The two additional terrestrial K-Pg deposition sites investigated in this study, Long Canyon and Seven Blackfoot Creek, show similar bulk $\delta^{34}$S values in the K-Pg boundary clay (4 ‰ and − 2 ‰, respectively), but only Long Canyon displays bulk S concentrations (~ $1500\,\mu g\,g^{-1}$) similar to those observed at the other terrestrial sites. The S concentration at Seven Blackfoot Creek (~ $500\,\mu g\,g^{-1}$) (Supplementary Table S2 in the SI) is likely too low to reflect vaporized anhydrite target deposition. The $\delta^{34}$S values for the Long Canyon site are similar to previously published[55] $\delta^{34}$S values for another Raton Basin terrestrial site, Sugarite (Raton Basin, New Mexico, USA, 2100 km from the Chicxulub impact structure), where an incomplete profile indicates that the $\delta^{34}$S values increase from 4.6 to 8.0 ‰ in the K-Pg boundary claystone in a similar manner to what is observed for the terrestrial sites discussed above.

In contrast to the terrestrial Tanis site, a positive peak in S and siderophile element (Co, Cr, and Ni) concentrations coincides with a negative $\delta^{34}$S peak for the marine Stevns Klint ($400–11,000\,\mu g\,g^{-1}$ and 18 ‰ to − 39 ‰, respectively) and Caravaca ($300–800\,\mu g\,g^{-1}$ and 19 ‰ to − 32 ‰, respectively) K-Pg site profiles (Fig. 2B, C, Supplementary Figs. S6, S7 and Supplementary Tables S2 and S4 in the SI). This is likely a result of S fractionation during microbial reduction of sulfate followed by pyrite sedimentation[48]. The Caravaca site additionally shows a positive $\delta^{34}$S peak in the limestone/marl section below the K-Pg claystone, which does not correspond to a positive peak in the S concentration, possibly indicating impact-related sulfate deposition, although the positive $\delta^{34}$S peak could also result from the incorporation of seawater sulfate (e.g., carbonate-associated sulfate in the limestone/marl sections[74]) at this marine site. In the case of both the Stevns Klint and Caravaca sites, a Ni/Cr ratio between 1.3–5.1, distinct from crustal values, is observed following the impact, which coincides with the increases in S concentration (Supplementary Figs. S6, S7, and Supplementary Table S4). Therefore, a portion of the S at these sites likely derives from airborne S related to the impact, which is also supported by previously published Ir data[48,68,75].

The S profiles obtained for the marine Brazos River site display positive peaks in S concentration coinciding with a more positive $\delta^{34}$S excursion ($6000–30,000\,\mu g\,g^{-1}$ and from − 40 ‰ to − 33 ‰), similarly as observed in previously published[52] S profiles for the distal marine Kawaruppu K-Pg boundary section (Tokachi District, Hokkaido, Japan, 11,000 km from the Chicxulub impact structure, $900–5400\,\mu g\,g^{-1}$ and from − 35 ‰ to 0 ‰[49]). In contrast to the other marine K-Pg boundary sites the S-profiles for these two sites are jagged (Fig. 2D and Fig. 1 in Kajiwara and Kaiho 1992[49]), indicating heterogenous S input in these K-Pg sediments. For the Brazos River site, the siderophile element

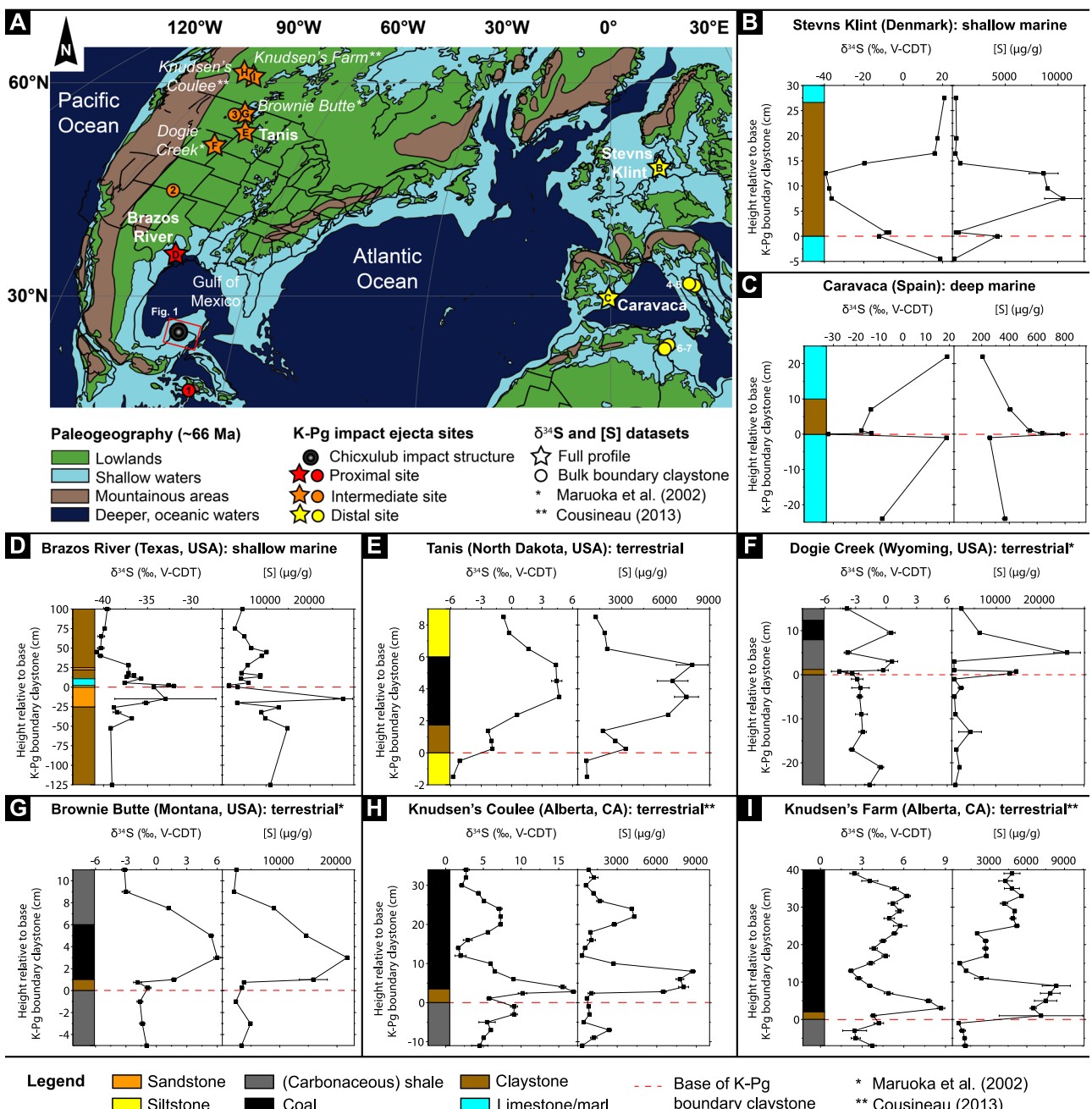

**Fig. 2 | Summarized S data for K-Pg boundary deposition sites. A** Simplified paleogeographic map reconstructed for the Late Cretaceous with K-Pg impact ejecta sites included in this study is highlighted. Different colors represent the proximity of the site to the Chicxulub impact structure, from 500 to 6700 km (modified from Goderis et al. 2021, Globally distributed iridium layer preserved within the Chicxulub impact structure. Sci Adv **7**, 1–13 (2021)[117]. © The Authors, some rights reserved; exclusive licensee AAAS. Distributed under a CC BY-NC 4.0 license http://creativecommons.org/licenses/by-nc/4.0/. Reprinted with permission from AAAS. The map published by Goderis et al. 2021[117] is redrawn after Paleoglobe for the Late Cretaceous, 66 million years ago, by C. R. Scotese, PALEOAMP Project 2012[122], and updated according to Snedden 2019[123], © Cambridge University Press. Reproduced with permission of C. R. Scotese and the Cambridge University Press through PLSclear, respectively). The stars represent K-Pg (Cretaceous-Paleogene) sites for which both bulk S concentration and δ³⁴S profiles around the K-Pg boundary were determined, while circles represent K-Pg sites for which the bulk S concentration and δ³⁴S value were determined in the K-Pg

boundary claystone only. The numbers correspond to K-Pg sites: (1) Beloc; (2) Long Canyon; (3) Seven Blackfoot Creek; (4) Frontale; (5) Fonte D'Olio; (6) Siliana; (7) Elles. **B–I** Simplified lithological profiles with corresponding measured bulk δ³⁴S and S concentration profiles for 8 K-Pg boundary sites, ranging from deep marine to terrestrial environments and from proximal to distal sites. These include: (**B**) Stevns Klint; (**C**) Caravaca; (**D**) Brazos River; (**E**) Tanis; (**F**) Dogie Creek (from Maruoka, T., Koeberl, C., Newton, J., Gilmour, I., and Bohor, B.F., 2002, Sulfur isotopic compositions across terrestrial Cretaceous–Tertiary boundary successions, in Koeberl, C., and MacLeod, K.G., eds., Catastrophic Events and Mass Extinctions: Impacts and Beyond: Geological Society of America Special Paper 356, p. 337–344, https://doi.org/10.1130/0-8137-2356-6.337[52].). **G** Brownie Butte (from Maruoka et al. 2002[52]). **H** Knudsen Coulee (from Cousineau 2013[56]). **I** Knudsen Farm (from Cousineau 2013[56]). Error bars represent the external uncertainty or 2 SD (based on digestion replicates and repeated measurements) and are often smaller than the markers. The dashed red line represents the base of the K-Pg boundary claystone or equivalent thereof.

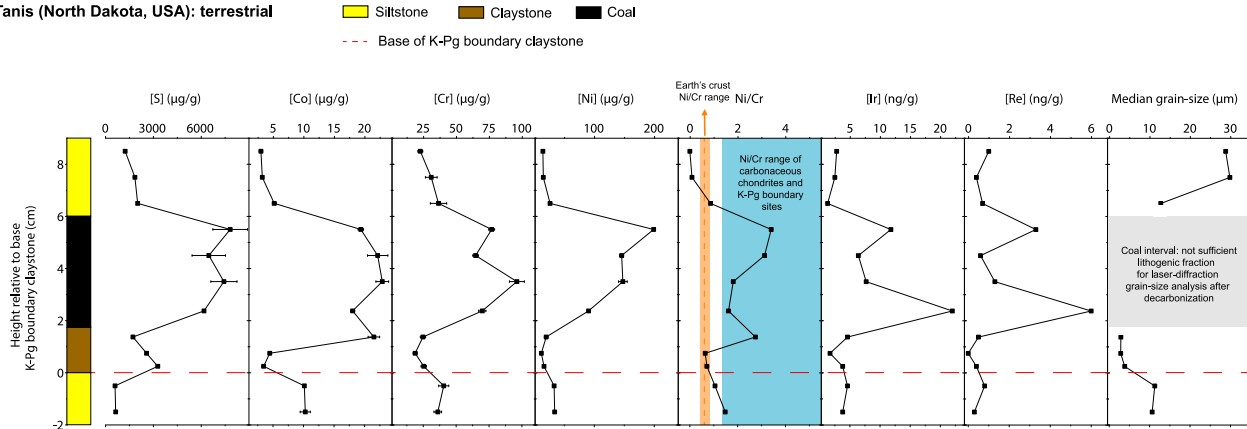

**Fig. 3 | Geochemistry and sedimentology of the Tanis K-Pg boundary site.** Bulk S, Co, Cr, and Ni concentrations; background subtracted Ni/Cr ratios; Ir and Re information values (concentrations close to the detection limit, DL, of the method); and previously published[13] median grain-size values of the Tanis K-Pg (Cretaceous-Paleogene) boundary site are presented. Error bars represent the standard deviation for two or more replicates and are often smaller than the symbols. The dashed red line indicates the K-Pg boundary claystone equivalent, based on sedimentological constraints, and the presence of microkrystites and shocked minerals[67]. Typical Ni/Cr values for the upper continental (UCC) crust[69] are shown with a dashed orange line, while the range is marked by an orange interval. The blue region indicates the previously published range for Ni/Cr values from other K-Pg boundary sites and for carbonaceous chondrites[68,70,71]. The gray region for the median grain size represents the Paleocene coal interval, where data are missing due to insufficient lithogenic fractions for accurate laser-diffraction grain-size analysis after decarbonization[13].

profiles (Co, Cr, and Ni) are also jagged (Supplementary Fig. S8 in the SI) and large variations are observed in the bulk S concentration and δ[34]S values for multiple sub-samples in the sandstone section (Fig. 2D), likely resulting from the admixture of S deposited from the atmosphere and, because of the proximity of Brazos River to the impact site, the wash-in of anhydrite-rich material derived from the crater region[76] (more detailed discussion found in the SI). A moderate Ir anomaly has previously been published for the Brazos River site[68,77–81], and the Ni/Cr values measured here range between 0.2 and 0.7 throughout the entire profile (Supplementary Fig. S8 and Supplementary Table S4), and are thus more consistent with continental crustal values (~ 0.5[68,69]). Together, these observations suggest strong dilution of the meteoritic component at this site likely due to impact tsunami/seiche waves.

The low δ[34]S values observed across the entire K-Pg boundary interval at the Stevns Klint (Fig. 2B), Brazos River (Fig. 2D), and Kawaruppu (Fig. 1 in Kajiwara and Kaiho 1992[49]) sites and in selected samples at the K-Pg boundary at the Caravaca site (Fig. 2C) may result from large S fractionation during microbial reduction, which has been suggested for these sites in previous studies[30,48,50,51]. The K-Pg sections in the mid-shelf marine sites, Stevns Klint and Brazos River are additionally marked by sedimentary enrichment of Mo, linked to blooms of stress-tolerant endobenthic foraminifera and indicating increased input of organic matter and hypoxic seafloor conditions at these sites following the K-Pg impact event[62]. Such low oxygen conditions are in line with the proposed microbial sulfate reduction mechanism and elevated input of poorly reactive organic substrates[82,83], leading to the post-impact formation of pyrite at these sites. From conventional marine sites, it is known that overall S isotope fractionation increases with increasing sulfate concentration above a critical level (reservoir effect[84]). At the deep marine Caravaca site, hypoxic conditions were brief[62], which also agrees with very low δ[34]S values measured only < 0.25 cm above the K-Pg boundary. Therefore, post-impact microbial S reduction leading to a large fractionation of the δ[34]S values is highly probable for these sites, meaning that the S concentration and δ[34]S signals determined at these sites no longer exclusively reflect deposition of atmospheric S.

Terrestrial environments, more specifically paludal to lacustrine settings, are considered more suitable for the investigation of impact-derived S as the background S concentrations are generally low and the deposited S less perturbed in relation to marine environments[52,85]. The impact-deposited sulfate at terrestrial sites may also undergo microbial reduction that could result in an underestimation of the total amount of impact-released S. However, these sites typically have lower sulfate concentrations, so S isotope fractionation is typically considered to be small[84]. Depending on the local environment, a substantial addition of sulfate to terrestrial environments, for example following seawater incursion events, has been observed to result in significantly higher S isotope fractionations. For example, large S isotope fractionation up to 50.5‰ has recently been observed in samples from a Cretaceous (~ 85 Myr ago) terrestrial inland rift basin located in northeast China[86]. At this Cretaceous terrestrial site, the large δ[34]S shift is attributed to singular or repeated seawater incursion events, during which the influx of sulfate-rich marine water into a sulfate-poor basin likely led to increased S isotope fractionation during microbial reduction. Comparably negatively correlated S concentrations and δ[34]S values are only observed briefly in the S profiles for one of the terrestrial sites included in this study (in the K-Pg boundary claystone of Dogie Creek, Fig. 2F), and attributed to microbial reduction during anoxic conditions in the Dogie Creek wetlands around the K-Pg boundary by Maruoka et al. 2002[52]. However, this negative δ[34]S shift is limited (1.7 ‰, from − 2.9 to − 4.6 ‰) and the rest of the Dogie Creek S profiles exclusively show positive correlations. In contrast to the marine site sediment profiles included in this study and the previously published terrestrial sediment profiles[86], microbial reduction fractionation effects are assumed to be small for all terrestrial sediment profiles discussed in this study (Tanis, Dogie Creek, Brownie Butte, Knudsen's Coulee, and Knudsen's Farm; Fig. 2E–I). All these terrestrial profiles display a comparable positive δ[34]S excursion, and none of the 5 sites show any type of highly negative δ[34]S values, despite deposition under strongly varying local environments in different basins. In addition, the paludal K-Pg boundary sites from the US Western Interior (Tanis, Dogie Creek, and Brownie Butte) are known for the preservation of a 1–3 cm thick boundary claystone that consists of a double layer microstratigraphy containing large quantities of different types of Chicxulub impact ejecta, yielding from base to top impact spherules, shocked mineral grains, and a clear iridium anomaly up to 16 ppb[68,87,88]. The unique preservation of this original ejecta chronology with a full platinum group element (PGE) profile at each of these terrestrial sites suggests that atmospheric

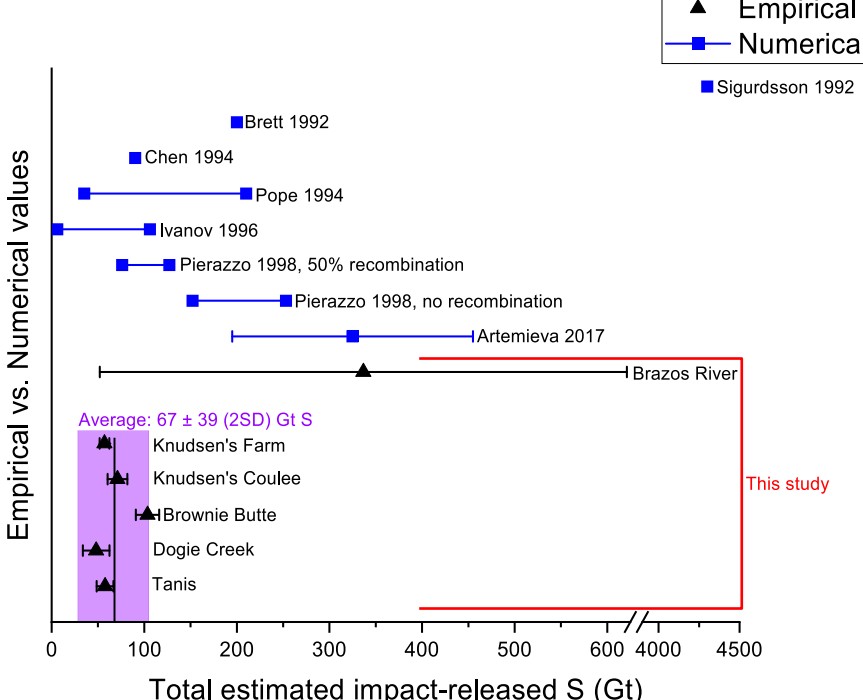

**Fig. 4 | Comparing estimates determined in current study to previously published values.** Comparison between empirical (black triangle markers) and numerical (blue square markers) estimations of the amount of S (in Gigatonnes; Gt) vaporized following the Chicxulub impact event. The black markers represent empirical S estimations based on $\delta^{34}$S and S-concentration values from various K-Pg (Cretaceous-Paleogene) boundary sites, with the error bars representing 2 SD

uncertainty of the values (based on digestion replicates and repeated measurements). The purple shadowed region shows the average of all these 5 K-Pg boundary sites (67 ± 39 Gt), excluding the Brazos River site due to the corresponding large uncertainties. Previously published numerical estimates (Sigurdsson et al. 1992[15], Brett 1992[34], Chen et al. 1994[35], Pope et al. 1994[22], Ivanov et al. 1996[36], Pierazzo et al. 1998[21], Artemieva et al. 2017[12], blue markers) are included.

processes are the dominant depositional mechanisms in the years to tens of years after impact.

## Estimated total amount of impact-released sulfur

The empirical estimate of the total amount of impact-vaporized S is calculated by combining large-scale isotope dilution and mass balance calculations, using the $\delta^{34}$S fingerprint of the target rock evaporite (18.5 ± 1.4‰ based on Yax-1, Y6, and UNAM-5-7), as well as S concentration and $\delta^{34}$S profiles in well-preserved terrestrial K-Pg boundary deposition sites (Supplementary Fig. S3 in the SI). The possible post-impact processes observed in the Stevns Klint, Caravaca, Brazos River, and Kawaruppu S profiles (Fig. 2B–D and Fig. 1 in Kajiwara and Kaiho 1992[49]) hamper the use of marine K-Pg profiles for accurate estimations of the amount of impact-released S into the atmosphere. Indeed, these processes can lead to incorrect estimations, as illustrated by the likely overestimated value for the Brazos River site provided in the current study (Fig. 4). Terrestrial K-Pg boundary sites are more likely to preserve the airborne S component and are more reliable for calculating the impact-released S. Moreover, many marine sites are stratigraphically condensed compared to more expanded terrestrial sites, providing only long-term estimates.

The empirical estimate of the amount of impact-released S is determined based on five terrestrial K-Pg boundary depositional profiles and is estimated to be 67 ± 39 Gigatonnes (Gt) (2 SD). These sites include Tanis (58 ± 9 Gt S, 2 SD), for which bulk S concentrations and $\delta^{34}$S values are obtained in this study, Dogie Creek[52] (48 ± 14 Gt S, 2 SD), Brownie Butte[52] (104 ± 13 Gt S, 2 SD), Knudsen's Coulee[56] (71 ± 11 Gt S, 2 SD), and Knudsen's Farm[56] (57 ± 5 Gt S, 2 SD), for which the estimates are calculated using previously published[52,56] bulk S concentrations and $\delta^{34}$S values. Although these five terrestrial sites are all located in North America within 3000–4300 km of the Chicxulub impact structure and may thus reflect a limited spread across the globe, the highly

reproducible S volumes obtained for these sites (Fig. 4) support their representativeness for the global non-ballistic atmospheric fallout of S-rich ejecta components. The highly reproducible impact-released S amounts (67 ± 39 Gt, 2 SD) of these 5 sites also suggest that the airborne S component is preserved with minimal influence of post-impact processes, such as significant S isotopic fractionation during microbial reduction of the sulfate or wash-out of the sulfate.

The new empirical estimate range of 28–106 Gt S is in excellent agreement with numerical estimate ranges published in the 1990s, particularly the data reported in Chen et al. 1994[35] (90 Gt S), Pope et al. 1994[22] (35–210 Gt S), Ivanov et al. 1996[36] (6–106 Gt S), and Pierazzo et al. 1998[21] (76–127 Gt S, assuming 50% recombination of vaporized S) (Fig. 4). However, the new value is a 5-fold lower relative to the more recent numerical estimate of 325 ± 130 Gt S by Artemieva et al. 2017[12] considered the most reliable estimate to date (Fig. 4).

One possible explanation for the lower estimates obtained in this study in comparison to the most recent modeling estimates is the occurrence of post-depositional effects, such as large negative S isotope fractionation during microbial reduction of sulfate to form pyrite, at the terrestrial K-Pg boundary sites. These post-depositional effects may locally affect the anhydrite signal and result in an underestimation of the true amount of deposited S in mass balance calculations. However, in this case, a large spread in the estimates for the different K-Pg boundary sites examined would be expected as post-depositional effects should vary for each site. In contrast, a close match is observed between the estimated amounts of impact-released S obtained for the five localities, rendering the possibility of a local overprint unlikely. A more probable explanation is that modeling approaches overestimated the atmospheric impact-derived load of S. Large remaining sources of uncertainties include, but are not restricted to, (i) underestimating the heterogeneity of the evaporites within the target rock, (ii) underestimating the critical shock pressures needed for degassing

and thus overestimating the percentage S degassed, and/or (iii) not considering recombination of S-bearing species within the impact plume. Firstly, the lack of anhydrite and gypsum in the most recent M0077A drill core, compared to previous drill cores (detailed discussion in the SI), likely illustrates a more heterogeneous distribution of evaporites within the impact structure than previously considered. To date, numerical simulations have mostly assumed a homogenous block of 22–63% anhydrite over the entire ~200 km impact structure. This is largely based on the distribution within the Yax-1 drill core[38–40], which may not be representative for the whole structure and may require additional constraints from future drilling campaigns. Therefore, it reflects a considerable source of uncertainty that must be considered in future simulations. Numerical estimations to date have assumed that all evaporite in the target rock is present in the form of anhydrite. However, if a fraction of the assumed anhydrite is gypsum, less S would have been produced by decomposition (~21% less per gypsum molecule), leading to a possible overestimation of impact-released S in the case of all numerical estimations. For more accurate future numerical estimations, the ratio between gypsum and anhydrite in the target rock should be better constrained. As discussed previously, critical shock pressure for degassing remains an important source of uncertainty that can cause extreme over- or underestimation of S release, as observed for Sigurdsson et al. 1992[15] (Table 1 and Fig. 4), and must be carefully considered for all numerical estimations. Artemieva et al. 2017[12] did take into account possible uncertainty related to shock pressures by running simulations with ±20 GPa compared to the normal runs, which changed the amount of impact-released S by ±35%. This uncertainty was included in the total error of the calculated estimate (±130 Gt S). However, this uncertainty may still reflect an underestimation if critical shock pressures differ by more than 20 GPa. Another important source of error is the occurrence of recombination effects, as previously observed for carbonates[89]. Based on clumped-isotope data and petrographic observations for carbonate-bearing samples in several drill cores from the Chicxulub impact structure, Kaskes et al. 2024[89] suggest that prior work likely overestimates the volume of impact-released $CO_2$ due to insufficient consideration of recombination effects. Consequently, the volume of impact-released S gases into the atmosphere and transferred globally is likely also overestimated by not considering the possible recombination of the S-bearing gases with CaO within the impact plume. The better agreement between the new empirical estimate of impact-released S obtained in the current study (67 ± 39 Gt S) and the numerical estimate range of 76–127 Gt S by Pierazzo et al. 1998[21], in which 50 % recombination of the S-bearing gases in the impact gas plume is assumed, compared to the estimate where no recombination is considered (152–253 Gt S), supports the importance of such possible recombination effects (Fig. 4). An additional source of error for the Artemieva et al. 2017[12] estimate is that no equations of state were used for anhydrite, simply because they are lacking. Today, the equations of state previously used by Ivanov et al. 1996[36] and Pierazzo et al. 1998[21] are considered unreliable, and an update is needed. Overall, several lines of evidence indicate that the Chicxulub cratering event released approximately 5-fold less S into the atmosphere than the most recent numerical estimate, with the amount of S released being more comparable to the values numerically estimated in the late 1990s.

The injection of ~70 Gt instead of 325 Gt of S, as advocated in this study, decreases the importance of the release of S in the killing mechanism, as S aerosols consequently contributed less to the drastic short-term climate perturbation as compared to the previous paleoclimate scenarios simulated by Tabor et al. 2020[24] and Senel et al. 2023[13] using the 325 Gt S estimate from Artemieva et al. 2017[12]. Tabor et al. 2020[24] proposed that the prime factor contributing to the initial extreme cold is soot, with the S agent playing a secondary role. Senel et al. 2023[13] instead suggest S as the main driver of the major cooling occurring during the first years of the impact winter, consistent with the finding of Brugger et al. 2017[28]. The other impact-released climate-active agents, such as soot and dust, affect the temperature to a smaller degree, however, act at longer time scales. Following the previously published paleoclimate modeling works, using the updated, lower S estimate, the extent and duration of the initial cooling spike should decrease even in the combined scenario of sulfur components, soot, and dust. The potentially milder impact winter scenario generated by the release of 67 ± 39 Gt (2 SD) S, compared to the most recent and reliable numerical estimate by Artemieva et al. 2017[12], allows for a crucial 'survival window' for many species[6,90–103], aiding in the persistence of at least 25% of species on Earth[4] during the K-Pg mass extinction.

## Methods
### Sample materials
All samples analyzed in this study are pulverized aliquots of samples used in prior studies. In previous publications reporting on these studies, field collection, removal of external debris, homogenization, and geophysical and geochemical characterization are explained in detail[9,38,42,44,58,62,68,104,105]. A full overview, including sample identification and lithological information, for all the samples characterized in this study, is provided in Supplementary Tables S1 and S2 in the SI.

### Bulk S concentration and S isotope ratio determination
Sample preparation and analysis—including digestion, chromatographic S isolation, S and siderophile element concentration determination, and S isotope ratio measurements—were carried out at the UGent-A&MS laboratory at Ghent University in Ghent, Belgium, and the ALS Scandinavia AB laboratory in Luleå, Sweden. All sample preparations at both labs were carried out in clean laboratory areas and followed the procedures in Rodiouchkina et al. 2023[106].

To minimize the waste of precious samples, preliminary S concentrations were obtained by non-destructive micro X-ray fluorescence spectrometry (μXRF) for all sample powders, and these concentrations were relied on to select the sample weights to be used for analysis. The μXRF measurements were carried out using an M4 Tornado benchtop μXRF surface scanner (Bruker Nano GmbH, Germany) with an Rh X-ray source at the AMGC Laboratory at the Vrije Universiteit Brussel in Brussels, Belgium[107,108].

For drill core samples that consisted of a large part of anhydrite (UNAM and Yax-1), sample amounts of ~10 mg were digested by adding 3 M HCl until no powder residue was observed. Aliquots of the sample digests were used for concentration determination. The digest was then diluted to 0.24 M HCl and loaded onto a conditioned cation exchange chromatography resin (Dowex 50W-X8), through which S, in the form of sulfate, passes, while the matrix cations are removed from the matrix by strong adsorption onto the resin. All other samples and matrix-matched elemental certified reference materials, soil (GBW07410), river sediment (NIST SRM 2704), and brick clay (NIST SRM 679), were digested using aqua regia (3:1 HCl:HNO₃) in closed beakers on a 110 °C hot plate for > 24 h. Aliquots of the standard and sample digests were used for concentration determination. The digests were evaporated at 70 °C and residuals were taken up in 0.24 M HCl before S was separated from matrix cations using the cation exchange protocol mentioned above. As many of these samples have a low S content (10 to 1000 μg g⁻¹) compared to matrix elements, the eluates were purified by submitting them to a second round of this cation exchange protocol, followed by an anion exchange chromatography protocol to separate S from residual matrix oxyanions. This protocol consisted of sulfate and other oxyanions adsorbing onto the anion exchange resin (AG-1-X8) and matrix cations passing through the column at 0.03 M HNO₃, followed by sulfate elution at 0.3 M HNO₃. Aliquots of all solutions were taken for S concentration determination after digestion, but also to ensure that the recovery of S was > 95% after chemical purification.

Determination of the concentrations of S and matrix elements was carried out using single-collector double-focusing inductively coupled plasma-sector field mass spectrometry (ICP-SFMS). Element XR (Thermo Fisher Scientific, Germany) instruments were used for this purpose at the UGent-A&MS laboratory and at the ALS laboratory. Sulfur isotope ratios were determined using multi-collector inductively coupled plasma-mass spectrometry (MC-ICP-MS) using a Neptune XT (Thermo Fisher Scientific, Germany) at the UGent-A&MS laboratory (determination of both $\delta^{34}$S and $\delta^{33}$S) and using a Neptune Plus, at the ALS laboratory (determination of $\delta^{34}$S only). In both cases, an Aridus II (Teledyne CETAC Technologies, USA) desolvating sample introduction system was used.

All S isotope ratios obtained ($^{34}$S/$^{32}$S and $^{33}$S/$^{32}$S) are presented using the $\delta$-notation according to Eq. 1, expressing deviations with respect to the international standard Vienna-Canyon Diablo Troilite (V-CDT) in per mil (‰). The given ‰ values are equivalent to mUr (milliUrey[86]). Correction for the bias caused by instrumental mass discrimination was accomplished using standard-sample bracketing and Si internal standardization[106]. Mass-independent anomalies are presented using the MIF tracer, $\Delta^{33}$S, which is calculated according to Eq. 2.

$$\delta^{3x}S = \left( \frac{\left( ^{3x}S/^{32}S \right)_{sample}}{\left( ^{3x}S/^{32}S \right)_{V-CDT}} - 1 \right) \times 1000 \qquad (1)$$

$$\Delta^{33}S = \delta^{33}S - 0.515 \times \delta^{34}S \qquad (2)$$

The accuracy of the $\delta^{34}$S values was assessed using International Atomic Energy Agency (IAEA) Ag$_2$S reference materials S1 ($\delta^{34}$S = − 0.30 ‰), S2 ($\delta^{34}$S = + 22.62 ‰), and S3 ($\delta^{34}$S = − 32.49 ‰) and both the $\delta^{34}$S and $\delta^{33}$S values using the recently produced $^{33}$S-enriched Na$_2$SO$_4$ standards[109] S-MIF-1 ($\delta^{34}$S = + 10.26 ‰, $\delta^{33}$S = + 14.81 ‰, $\Delta^{33}$S = + 9.54 ‰) and S-MIF-2 ($\delta^{34}$S = + 21.53 ‰, $\delta^{33}$S = + 22.42 ‰, $\Delta^{33}$S = + 11.39 ‰). The IAEA standards were digested following the procedure described by Craddock et al. 2008[110] and Rodiouchkina et al. 2023[106], according to which 5 mL of 7 M HNO$_3$ was added to the standards and the mixture was left to evaporate to dryness on a 70 °C hot plate. The dry residue was further digested using 3 mL of concentrated HNO$_3$ and the resulting Ag in solution was precipitated using 2 mL of 6 M HCl before taking the mixture to dryness on a 70 °C hot plate again. When dry, the residuals were taken up with 0.24 M HCl and chemically purified in the same way as the samples. The MIF standards were dissolved using 0.24 M HCl and S was isolated using the same chemical purification method as used for the samples.

The expanded uncertainty ($U$) is calculated using the sum square approach[111–113] according to Eq. (3), where $k$ is the coverage factor (here, $k$ = 2, confidence level of approximately 95%) and $u_c$ is the combined uncertainty of the internal precision of one measurement ($SD_{int}$), the within-session repeatability ($SD_{within}$), the between-session repeatability ($SD_{between}$), and the repeatability between separate sample preparations ($SD_{sample\_prep}$). These were estimated using matrix-matched standards and by digesting at least one sample of each lithology more than once.

$$U = k \times u_c = 2 \times \sqrt{SD_{int}^2 + SD_{within}^2 + SD_{between}^2 + SD_{sample\_prep}^2} \qquad (3)$$

## Total reduced inorganic S and sulfide isotope ratio determination

Chromium-reducible sulfur (CRS; essentially pyrite; FeS$_2$), was extracted with hot acidic Cr(II)chloride solution (Fossing and Jørgensen, 1989[114]). The released H$_2$S was precipitated as ZnS and then total reduced inorganic S (TRIS) was determined spectrophotometrically

(Specord 40 spectrophotometer) following the methylene blue method[115]. For sulfur isotope ratio ($^{34}$S/$^{32}$S) determination, the ZnS was converted to Ag$_2$S by the addition of 0.1 M AgNO$_3$ solution with subsequent filtration, washing, and drying of the Ag$_2$S precipitate. Isotope ratio measurements were carried out using CirmMS with a Thermo elemental analyzer connected to a Thermo Finnigan MAT 253 gas isotope ratio mass spectrometer via a Thermo ConFlo IV split interface in the BGC lab at IOW. IAEA-S1, -S2, -S3, and NBS127 isotopic reference materials were used to calibrate the mass spectrometric signals[116].

## Calculation of the estimated amount of impact-vaporized S

The estimated amount of impact-vaporized S is calculated by combining traditional isotope dilution and mass balance calculations. In the post-impact winter hypothesis, S-species injected into the atmosphere following the Chicxulub impact event would first distribute globally, causing cooling and darkness before they would gradually return to the Earth's surface via dry or wet deposition processes. In bedrock with naturally low S content and a $\delta^{34}$S value significantly different from that of the sulfate aerosols related to vaporized target anhydrite, the global event is recorded in the impact event deposit sediments as an S offset in both the S concentration and the $\delta^{34}$S value. The thickness of the post-impact event deposits is dependent on the sedimentation rate, but also on the state of preservation of the record. To determine the amount of S released during the Chicxulub impact event, sediment profiles of K-Pg boundary sites outside of the impact crater were therefore investigated for possible positive offsets due to deposition of the ejected S. The K-Pg boundary sites selected all contain the classic ejecta layer, comprising markers such as impact spherules (microtektites and/or microkrystites), shocked quartz and the positive iridium anomaly[62,68,117].

For example, a clear positive spike in both the S concentration and the $\delta^{34}$S value is observed at the terrestrial Tanis (North Dakota, USA) site (Fig. 2E and Supplementary Fig. S3 in the SI). We assume that impact-deposited S is solely responsible for this offset and that the S concentration and $\delta^{34}$S value in the sediment profile before this spike represent the background values for this site, not attributable to atmospheric S deposition. These background values are, therefore, subtracted to obtain the S concentration (Eq. 4) and $\delta^{34}$S value (Eq. 5) stemming from post-impact atmospheric deposition at the K-Pg boundary sites. The amount of $^{34}$S-enriched impact event deposition in each sediment sample at the K-Pg boundary sites ($C_{Deposit\ per\ sample}$) is then calculated for each profile interval by multiplying the background-corrected offset S concentration ($C_{K-Pg\_site\_deposit}$) with the ratio of the background-corrected $\delta^{34}$S value related to the K-Pg boundary site ($\delta^{34}S_{K-Pg\_site\_deposit}$) to that of the target anhydrites ($\delta^{34}S_{Impact-target}$) according to Eq. 6. This amount ($C_{Deposit\ per\ sample}$) is then multiplied by the density determined gravimetrically (or average tabulated values taken for each lithology type) at each sample point in the profile ($\rho_{For\ each\ sample\ in\ profile}$) and subsequently integrated across the entire profile using the vertical thickness of each sample unit ($VL_{Per\ sample}$) to obtain the amount of impact-deposited S collected over the entire K-Pg site profile ($M_{Deposit\ over\ entire\ profile}$) according to Eq. 7. The total amount of impact-vaporized S is calculated by extrapolating this value to the entire surface of the Earth ($A_{Earth}$ = 510,000,000 km$^2$), assuming equal global deposition, according to Eq. 8. The reported uncertainty on this value is expressed as 2 SD and as expanded uncertainties for the $\delta^{34}$S values for the target anhydrite as well as for the measured S concentration and $\delta^{34}$S value of the sample points in the distal site profile.

$$C_{K-Pg\_site\_deposit} = C_{K-Pg\_site\_total} - C_{K-Pg\_site\_background} \qquad (4)$$

$$\delta^{34}S_{K-Pg\_site\_deposit} = \delta^{34}S_{K-Pg\_site\_total} - \delta^{34}S_{K-Pg\_site\_background} \qquad (5)$$

$$C_{Deposit\ per\ sample} = C_{K-Pg\_site\_deposit} \times \frac{\delta^{34}S_{K-Pg\_site\_deposit}}{\delta^{34}S_{Impact-target}} \quad (6)$$

$$M_{Deposit\ over\ entire\ profile} = \sum (C_{Deposit\ per\ sample} \times \rho_{For\ each\ sample\ in\ profile} \times VL_{Per\ sample}) \quad (7)$$

$$M_{Total\ imact-released} = M_{Deposit\ over\ entire\ profile} \times A_{Earth} \quad (8)$$

## Data availability

The elemental concentration (S and siderophile elements) and isotopic composition ($\delta^{34}$S, $\delta^{33}$S, and $\Delta^{33}$S) data generated in this study have been deposited in the Figshare data repository[118]. Further, bulk S concentrations, $\delta^{34}$S, $\delta^{33}$S, and $\Delta^{33}$S values are presented in Supplementary Tables S1 and S2 in the supplementary information file. Supplementary Table S3 in the supplementary information, presents the total reduced inorganic S (TRIS) and sulfide-specific $\delta^{34}$S. Siderophile element concentrations and median grain size are presented in Supplementary Table S4 in the supplementary information. All powdered selected lithological units of the drill cores within and around the Chicxulub impact structure (PEMEX Y6, UNAM-5, UNAM-6, UNAM-7, ICDP Yax-1, and IODP-ICDP Expedition 364 M0077A), as well as all profiles and bulk samples taken from K-Pg boundary deposition sites (Tanis, Stevns Klint, Caravaca, and Brazos River) measured in this study were collected in previous studies[9,38,42,44,58,62,68,104,105]. Requests for samples should be sent to S.G., J.V., P.K., and P.C.

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

## Acknowledgements

The authors acknowledge the Research Foundation Flanders (FWO) for financial support for this study through the EOS-Excellence of Science program (ET-HoME—ID 30442502). FWO is also acknowledged for providing the funding for the acquisition of MC-ICP-MS instrumentation for the A&MS laboratory (ZW15-02—G0H6216N). S.G. and P.C. acknowledge support from the VUB Strategic Research Program, FWO, and the Belgian Science Policy Office (BELSPO). C.B.S. is supported by grant 12AM624N of the FWO. P.K. was financially supported by FWO PhD fellowship 11E6621N. Ö.K. was supported by the BELSPO through the Chicxulub BRAIN-be (Belgian Research Action through Interdisciplinary Networks) project. J.V. is funded by the Belgian Science Policy Office (BELSPO) through the FED-tWIN project MicroPAST (Prf-2020-038). We sincerely thank ALS Scandinavia AB for participating in this study, giving K.R. permission to use their facilities, and for the invaluable technical support. Robert DePalma is thanked for providing access to the Tanis K-Pg site and assistance in sediment sampling. The authors also extend thanks to Natalia Artemieva for valuable input regarding the merits and pitfalls of simulated estimates.

## Author contributions

K.R. led the writing of the manuscript in close partnership with S.G. S.G. was responsible for the conceptualization, selection of the samples of interest, and main supervision of the study. F.V. contributed to the supervision of the study and was responsible for the project administration and funding acquisition. J.V., S.G., and P.K. provided the samples. K.R carried out the sample preparation, analysis, and data treatment of the bulk S elemental/isotopic determinations with aid from I.R. K.R created Fig. 4 with input from S.G., Ö.K., and P.C. P.K provided μXRF S concentrations, assisted with literature study, and created Figs. 1– 3 and Supplementary Figs. S1, S6–S8. M.E.B. contributed with TRIS analysis and contributed with pyrite S isotopic analysis of selected samples. S.G., C.B.S., P.K., Ö.K., M.E.B., I.R., J.V., P.C., and F.V. contributed to reviewing and editing the manuscript. All authors approved the final draft of the manuscript.

## Funding

## Competing interests

The authors declare no competing interests.
