## [Transparent Peer Review file · Nature Communications]

Reduced contribution of sulfur to the mass extinction associated with the Chicxulub impact event

Corresponding Author: Dr Katerina Rodiouchkina

Version 0:

Reviewer comments:

Reviewer #1

(Remarks to the Author)

The manuscript by Rodiouchkina et al. describes an investigation of elemental sulfur and sulfur isotopes across the K-Pg boundary in marine sections from the impact crater, continental shelf and slope, and deep ocean as well as terrestrial sections from the continental US. The results are used to interpret the amount of sulfate aerosols that were formed during and shortly after the impact and their climatic and biotic impact compared to soot from wildfires and dust from pulverization of the impact terrane. The results suggest that sulfate aerosols were less voluminous than several recent studies. This is important as it provides a more accurate account of the climatic after-effects of the impact and the time frame for cessation of photosynthesis which was the likely driver of mass extinction. The authors use the results to simulate the post-impact cooling and this indicates that the initial pulse was not as dramatic as previous interpretations suggesting that the extinction took longer.

The approach is not novel in that previous studies have attempted to model both sulfur and sulfur isotope concentrations, but what the study does do is provide a more expansive dataset as well as give a potentially more holistic interpretation of the compiled data.

Overall, I believe that this manuscript is provocative and of wide interest to the community and the broader public. The issue of dinosaur extinction has substantial appeal, and this alone will fuel interest in the manuscript. However, there is one major issue that I believe needs to be resolved before the manuscript can be considered further. The key climatic events of the K-Pg impact happened so rapidly that they are often difficult to resolve in distal deep-sea sections that were deposited very slowly but offer the simplest stratigraphic interpretation. In sections that were deposited closer to the crater, key events can be more readily detected, the crucial issue is deciphering how materials were delivered to the land surface. The bulk of these thick sections were delivered by high energy processes like tsunami and seiche waves which distort primary geochemical and climatic signals through redeposition. The key in interpreting data like the sulfur data presented here is deciphering the signals that were derived via airfall because these are the true signals that have climatic significance. And in short, I am unconvinced that the samples that are key to the interpretations made here were indeed airfall, or more precisely entirely modulated by airfall.

The authors explain the highly negative sulfur isotope signals at the shelf and deep-sea sites as a result of sulfate reduction, though the Brazos section contains a positive anomaly which is different from the other sites. The authors interpret this as a combination of a mixture of deposition from airfall and redeposition likely from tsunami and seiche waves that traveled to the shelf. At the same time the authors interpret the sulfur data from the Tanis and other western Interior sections in terms of primary airfall when most of the Tanis section at least is thought to be deposited by seiche waves as described in the initial DePalma et al (2019) publication. DePalma et al. did interpret several horizons to be primary airfall but that interpretation was not entirely convincing to me. As far as I know the other western interior sections have not been confirmed in detail to be airfall either. The authors state in line 306 that terrestrial sections are likely to be only influenced by the airborne component and are therefore more reliable. However, I have not seen any proof of this here or elsewhere. I wonder whether the signals reported from Tanis and the other sections could represent mixing and dilution of the atmospheric signal by the seiche waves. This is so central to the overall interpretation that I do not think the manuscript can go forward without a more convincing interpretation of the mode of deposition in the Tanis and other terrestrial sections. Note that I have the same issue with the recent Senel et al. (2023) paper on silicate dust from the Tanis section.

If most of the sediment in the terrestrial sections was delivered by seiche waves, the question is where did it come from? Specifically, where did its sulfur come from? Was it contemporaneous sediment containing sulfate aerosol fallout from the impact (ie airfall) redeposited locally, or was the sulfur derived from other sources including older sediments or exposed bedrock, or was it a mixture of the two sources. Bottom line is that there needs to be a discussion of the significance of the isotope signals with respect to the mode of deposition and a more convincing proof that the signals are indeed airfall.

The authors have also collected data from sites within the crater itself to determine the sulfur chemistry of the target. The results show great heterogeneity which is to be expected. One thing that I missed however was the interpretation of the IODP site M077. This site contains both pre and post impact rocks and I'm a little confused whether there is any significance of the post impact record. The data in Figure 1B aren't plotted at high enough resolution to determine this.

Finally, I wonder if the authors should be presenting data (Figure 2B) that have little significance for the study for example the data from Caravaca and Stevns Klint. I would prefer to see the data from the other terrestrial sections even if they are from other sources.

Minor points

Line 68 also carbon from buried petroleum (Kaiho et al. and Lyons et al. 2020)

Line 87-90 need references for the models.

Line 265 don't you mean a positive isotope anomaly?

Line 273, repetitive with sites named, not necessary

Sulfur isotopes and concs in Figure 2 needs Ir data for airfall

Figure S2 should have labels on to indicate which section is which

Reviewer #3

(Remarks to the Author)

Review of "Reduced contribution of sulfur on the mass extinction associated with the Chicxulub impact event" by Rodiouchkina et al.

This paper uses measurements of sulfur (S) concentrations and isotopic compositions from terrestrial sections across the K-Pg boundary to estimate the emissions of S-bearing gases from the Chicxulub impact associated with the end-Cretaceous mass extinction. Since the amount of S-bearing gases from the impact is difficult to quantify from geophysical simulations of the impact process, but highly relevant for the climatic effect of the impact, this paper adds an important piece of information required for a better understanding of the end-Cretaceous extinction event and definitely warrants publication.

That being said, there are severe issues with the climate modelling part of the paper which imply that the manuscript cannot be published in its present form and requires major revision.

Major comments:

(1) In several places, the model is described as a "General Circulation Model (GCM)" (e.g. in lines 162-163, 753). By today's standards, this would imply an AOGCM representing the full three-dimensional dynamics of both the atmosphere and the oceans. In this case, however, the model is "just" an atmospheric general circulation model (AGCM) coupled to a strongly simplified ocean model which will affect the results (see below). This should be mentioned and discussed in the main text of the paper, not just in the methods section. Also, I would not call this model "state-of-the-art" (line 162) since it is not an AOGCM.

(2) Even in the methods section, there is a certain lack of details regarding the model. How thick is the mixed layer in this model? How is meridional heat transport in the ocean prescribed since it obviously cannot be simulated dynamically? What about the sea ice model, does it capture the dynamics of sea ice or just the thermodynamics? Again, this is important to be able to judge the reliability of the climate modelling results presented in the paper.

(3) I think something might be wrong with the "global surface temperature" maps presented in Fig. 5. I am particularly worried about the relatively high temperatures in the polar oceans after the impact. I suspect that the temperatures could be surface rather than surface *air* temperatures which would explain why the polar oceans (which should be covered by sea ice) are only at temperatures at or above the freezing point of seawater rather than at temperatures similar to continental areas (which is what I would expect for cold air above sea ice in polar regions). If the temperatures shown in the maps are indeed surface *air* temperatures, something could be wrong with the treatment of sea ice in the model. In any case this needs to be double-checked and/or discussed, in particular with respect to the emphasis on global mean temperatures being above the freezing point throughout the paper (e.g. lines 55-56, 370-372, 388-389, Fig. 6).

Please note that other model studies present surface *air* temperatures, of course, so this is relevant for the comparison with other studies and proxies - and for the sweeping interpretations in the paper regarding species survival. By the way, the continental areas appear to be at temperatures below freezing even for this model study, so I fail to see how the results presented in this paper would lead to significantly different conclusions regarding species survival compared to other studies.

Finally, are there really regions with annual mean surface (air?) temperatures beyond 40 degrees for a 560 ppm pre-impact state? This would be really surprising - and add to my suspicion that the limitations of the model might be more significant than the authors care to admit... These extremely high sea surface temperatures in the tropics could be due to a lack of meridional heat transport maybe?

(4) The lack of a full ocean model (and I suspect also of a dynamics sea-ice model) will certainly affect the results. For example, in this model study the lowest temperatures are reached relatively early, and the recovery is faster compared to studies using full ocean models (e.g. lines 360-361, 364-365). This is most likely due to a lack of the full thermal inertia of the ocean in model. The influence of model limitations on the results definitely needs to be discussed.

Minor comments:

Fig. 5: I would also suggest to use a colour scale in which the freezing point is white rather than blueish to make it easier to see which regions are below freezing in the figure.

The discussion in lines 397-409 focusses solely on Senel et al. (2023) but should include other relevant studies, in particular Tabor et al. (2020), but also Brugger et al. (2017).

Reviewer #4

(Remarks to the Author)

In this study, Rodiouchkina and colleagues report S-isotopic data and sulfur contents from various terrestrial and terrestrial-marine sections across the K-Pg boundary. The authors developed a simple mass balance model to estimate the amount of impact-vaporized sulfur. The S-isotopes and sulfur contents are two major variables in their model calculations. The authors argue that their estimate of vaporized sulfur from the Chicxulub impact is approximately five times lower than that proposed by Artemieva et al. (2017), but it aligns with numerical estimates from the 1990s. Subsequently, the authors utilized the same GCM model they published last year (Senel et al., 2023; ref. 12) to discuss the cooling effect caused by sulfate aerosols and their implications for the demise of the non-avian dinosaurs. The geochemical measurements are in high precision and the paper is well written. The results presented in this study contribute a new piece of evidence for our understanding of how dramatic climate change can lead to mass extinction in the geological past and potentially in the future. I have a few major comments which may be helpful when the authors revise their manuscript:

1: S-isotope analysis constitutes a significant component of your presentation. However, S-isotopes are nearly absent from your introduction (lines 59-163). I encourage the authors to enlighten readers on how S-isotopes can assist in estimating vaporized sulfur from the Chicxulub impact. Highlighting the novelty of your empirical estimate could also provide valuable context for the readers. If you find the introduction too lengthy, consider condensing the summary of previous work (lines 91-117) for brevity.

2: I agree with the authors that a positive peak in sulfur concentration, along with a positive $\delta^{34}\text{S}$ value, may be a primary signal resulting from the impact. I am also convinced that the negative $\delta^{34}\text{S}$, coupled with an increase in sulfur contents from Stevns Klint, Caravaca, and Brazos River, does not exclusively result from atmospheric sulfur deposition. It is correct to suggest that 'This is likely a result of S fractionation during microbial reduction of sulfate followed by pyrite sedimentation' (lines 260-261). However, it is important to note that sulfate deposited from the atmosphere in rivers and lakes can also undergo biological reduction to form pyrite.

I suggest the authors to provide a more detailed explanation of their arguments regarding the origin of the negative $\delta^{34}\text{S}$ shift. Several factors can influence the S-isotopic compositions of biogenic pyrite, and it seems that an increase in sulfate concentration from seawater input is a plausible candidate, if not the only one (see a recent study by Xu et al., 2023, *Chemical Geology* 642, 121790). Readers would benefit from a more comprehensive interpretation in addition to citing previous work.

3: I suggest that the authors elaborate on their model construction in the section from lines 298-306. While it appears concise, this aspect is a crucial contribution to your study and could benefit from a more detailed explanation.

4: As I'm not a GCM model person, I am unable to provide comments on the validity of the model and the simulated results. I presume that other reviewers with expertise in this area can offer valuable insights.

Reviewer: Yanan Shen

Version 1:

Reviewer comments:

Reviewer #1

(Remarks to the Author)

This is the second time I have reviewed this manuscript and I'm incorporating the author's detailed comments on my first review in my analysis. Overall, I believe the authors have done a very good job of addressing my previous concerns. This is a very extensive piece of work both in the number of analysis, and the depth and extent of the interpretation, so it's really hard to go justice to all of the science in a short format manuscript. I think that is what gave rise to my previous concerns and misunderstandings and that is what might still drive some of them.

In summary, in my mind this is a very impressive and important piece of science that deserves to be published. I'm more convinced the interpretation of S volume is realistic than I was before. I do have some doubts about the Tanis section as I describe below, and I think the authors could continue to resolve some of them, but it seems that there is somewhat less focus on Tanis here than in previous version which may just be a perception. Overall, I support publication of this provocative paper with some minor revisions. As I said before I think that this novel interpretation of the role of sulfur gasses will drive discussion on extinction mechanisms and recovery in a positive direction.

The authors have presented so much data and again I congratulate them on this. They cover a lot of ground in a relatively short manuscript including the detailed sulfur isotope geochemistry and climate modeling based on it that are central to their main argument. Their data and interpretation therefore spill over into a lengthy supplement. This concern is not unique to this manuscript, its true of many papers that end up in short format journals. Being honest I just skimmed the supplement for information on Tanis and this brings me back to one of my other concerns in the first review. There are many possible ways of telling this story admittedly and I hope my suggestions below will make it clearer.

The authors have responded to my concerns that the Tanis section includes redeposition from seiche deposits in addition to airfall deposits. They rely on the Ir and other platinum metal concentrations to make the case for airfall in the absence of detailed sedimentology (I'm assuming the palynology they refer to here is published). In addition, the authors have added some text (Lines 331-354) on the Western Interior alluvial and lacustrine terrestrial sections in general and the possibility of sulfate reduction in them. I really appreciate this but I suggest they give more detail on the background of what is driving the interpretations of the sedimentology of these sections as a group. Much of the audience like myself are unfamiliar with this literature and I think it would help the manuscript enormously. This is relevant because presumably the coal unit must have been deposited over many years and not just from airfall (lines 281-283).

The sedimentology of Tanis itself has not been investigated in detail and given the importance of the interpretations that are being made from this section here and elsewhere a detailed investigation of process sedimentology really needs to happen. I'm not saying this should be in the scope of the current study though. The authors themselves have not been able to document the sedimentology in further detail because they could not analyze grain size in the key coal interval (Figure S6). Down the road there needs to be more detailed sedimentology.

Finally given the importance of Tanis to the overall interpretation, I think it would be more convincing to have a stand-alone figure in the main text on this section, in particular, including the Ir and other REE data in the main text not the supplement.

I've spent some time looking at the Junium et al. paper that focused on Brazos and that was one of my previous concerns. I now agree with the authors' interpretation of this dataset, the key sulfur isotopes interpreted by Junium were from the tsunami interval and thus likely a combination of sources, (largely) transported in from the crater as well as (minor) from airfall. So I agree with the interpretation that Junium et al. have overestimated the volume of sulfate aerosols. I do think that there needs to be more discussion about the deposition of the post tsunami upper units at Brazos (E, F and G) in Figure and how the S values differ from the tsunami beds. In addition, I think that the authors need to make it clearer how their Brazos analyses differ from those of Junium et al.

Minor suggestions

Line 121, "risking possible overestimation" is unclear. Does this mean the authors didn't consider these factors or the fact that they didn't risked overestimation?

Line 151 Do you mean the impact crater itself?

Lines 171-174 The statement about Brazos being a tsunami deposit depends on where Junium et al.'s measurements are made since some of the record is fallout from suspension and storm deposits. So they need to clarify that Junium et al.'s measurements are from the tsunami interval at Brazos.

Line 252 Should have a connection with previous section for example say that comparing global boundary sites with the target sulfur.

Lines 253-263 Should give figure numbers for the individual sections (ie. Fig 2B-2E)

Line 262 Weren't some of these measured in previous pubs?

Line 259 "sampled in the same region as the one previously used" this is unclear.

Line 280 Don't you mean Figure S6 for grain size?

Lines 322-326 Recommend this sentence be divided in two.

Line 329 "cannot be ruled out" is a little weaker than previous text in the paragraph which implies sulfate reduction fairly strongly.

Line 337 Addition of sulfate from where? Seawater incursion?

Line 342 "the large $\delta^{34}\text{S}$ shift" what does this refer to? The K-Pg or other impacts?

Line 374-376 Also these sites are condensed so only give very long-term estimates, terrestrial sites are more expanded

Line 399 Be specific what the post depositional effects are.

Line 460-477 This is more important. This paragraph should be reversed with following paragraph that specifies that soot and silicate dust are smaller drivers than sulfur. Without this context the impact of sulfur on its own is confusing. There needs to be one stand alone paragraph focusing on the implications on life at the end of the manuscript.

Figures

Figure 2 need to plot S concentrations in all of the panels at the same scale to make the changes more comparable. I would suggest this for the isotopes too because it would be more graphic to have the positive and negative shifts going in different directions even though it may be hard. Also, the red dashed line is the base of the K-Pg boundary section.

Reviewer #3

(Remarks to the Author)

Review of the revised version of "Reduced contribution of sulfur to the mass extinction associated with the Chicxulub impact event" by Rodiouchkina et al.

The authors have now taken into account most of my comments on the climate-modelling part of the paper. However, the additional details in the model description only confirm my suspicion that the climate-model setup used for this study is simply unsuitable for investigating the climate of the late Cretaceous and, more importantly, the impact scenarios. I will detail my major concerns in the comments below. As it stands, the manuscript cannot be published with the modelling part included, in my opinion, or the climate modelling has to be repeated with a vastly improved model setup. Otherwise no-one in the climate modelling community would believe these results.

Also, I see from the other reviews that there is quite a bit of a discussion concerning the empirical part of the manuscript on which I cannot provide any expertise.

Major comments:

The climate model is not suitable for the current investigation. This statement needs some background information: State-of-the-art coupled climate models (AOGCMs) combine a 3D atmosphere model (AGCM) with a 3D ocean model (OGCM) including a sea-ice model which captures both sea-ice dynamics and thermodynamics. As the authors point out correctly, these models are computationally expensive, so faster, simplified models are frequently used in paleoclimate studies. This is perfectly alright as long as the simplifications do not severely affect processes relevant for the study, and as long as the limitations are taken into account when discussing the results. Neither is the case for this manuscript, unfortunately.

A typical model used for paleoclimate modelling could be an AGCM (as in the case of this study) coupled to a simplified ocean and sea-ice model, for example a mixed-layer ocean model with prescribed meridional heat transport and a thermodynamic sea-ice model. This is not ideal, of course, because the meridional heat transport in the ocean cannot adjust to changes in climate in this case, and because sea-ice dynamics is important (in addition to thermodynamics), but it would be acceptable for this study.

The problem with the model setup used here is that it is not just simplified with respect to state-of-the-art climate models, but even overly simplified compared to simplified setups typically used in paleoclimate studies. There are two substantial shortcomings of the model setup used in the manuscript:

(1) There is no meridional heat transport in the ocean at all which is most likely the reason behind the unrealistically high tropical sea-surface temperatures (see also my comment on the original version of the manuscript) rather than the unconvincing explanation in the rebuttal. This will yield a physically unrealistic pre-impact climate state and a completely unrealistic response of the ocean (and the climate) in the impact scenario, making the simulations in essence meaningless.

(2) There is no sea-ice model at all, not even a simple thermodynamic model. This is a crippling limitation. First of all, I do not believe the authors' argument that there won't be any (even seasonal) sea-ice at 560 ppm and for a solar constant which is lower than today. Of course there will be sea ice (at least in winter) even in the warmer pre-impact state! Much worse, the drastic global cooling after the impact cannot be meaningfully simulated without taking sea-ice formation into account. Unfortunately, this makes the entire modelling effort rather pointless from a climate physics point of view. This is also the reason for the very warm polar regions (see also my comment on the original version of the manuscript) rather than the argument given in the rebuttal. By the way, the absence of a sea-ice model also explains why global surface air and surface temperatures are so similar. They would not be in reality since the air above sea ice gets really, really cold...

Assuming that the empirical part of the manuscript is sound, there are two ways to move forward. Either the authors remove the climate-modelling part from the manuscript since it is unsuitable. Or the authors redo all the simulations with a model setup which includes (prescribed) meridional heat transport in the ocean and (at least) a thermodynamic sea-ice model.

Note that this is not nitpicking. The limitations of the model setup used in the study are really fundamental and not acceptable within the paleoclimate modelling community.

The other thing which worries me is that this is not the first study by the authors on the Chicxulub impact with this model setup. In my opinion, the model limitations are so severe that this reaches the point where one could think of submitting clarifying corrigenda/addenda for the earlier publications...

Additional minor comments:

I. 474: I appreciate that the authors have replaced "global temperature" by "global surface temperature" throughout but I think that it should be made even clearer that these are not surface air temperatures, e.g. by inserting "(not surface air)" or similar when temperatures are first mentioned.

I. 484-486, "No value within this new empirical estimated range of impact-released S gives rise to global-average temperatures below freezing point (Fig. 6).": I am not sure why this is emphasised. Also, this is so severely affected by the model limitations that the statement is not supported by the simulations presented in the manuscript.

Reviewer #4

(Remarks to the Author)

I have carefully evaluated the revised manuscript. The authors have addressed my comments satisfactorily. Therefore, I recommend it for publication.

Yanan Shen

Response to the reviewers - Nature Communication manuscript NCOMMS-23-53601:

“Reduced contribution of sulfur to the mass extinction associated with the Chicxulub impact event”

Katerina Rodiouchkina, Steven Goderis, Cem Berk Senel, Pim Kaskes, Özgür Karatekin, Orkun Temel, Michael Ernst Böttcher, Ilia Rodushkin, Johan Vellekoop, Philippe Claeys, Frank Vanhaecke

Dear editor and reviewers,

First of all, we would like to thank you for your time and effort reading through this manuscript and for your constructive comments and remarks. Please find the revised PDF document with all changes based on your comments and suggestions marked through track changes. Below, a summary of implemented changes are provided followed by a point-by-point response to each comment is provided as requested. We hope that our responses fully address the comments received during the reviewing process. The response from the authors is highlighted in the color blue.

Yours sincerely,

Katerina Rodiouchkina, on behalf of all authors

Summary of implemented changes:

Graphical abstract; Fig. 1, 2, 4, 5, 6; and Fig. S3, S10-15 were updated.

Table S4; Fig. S1, S5-9, S16-17 were added to the supplementary information.

Several paragraphs and sentences were added to the Introduction (Lines 153-159, 163-171, 177-180), Result and discussion (Lines 229-234, 262-263, 278-282, 292-296, 297, 307-310, 311, 322-326, 331-354, 390-393, 483-492), Materials and methods (Lines 770-774, 885-887 and 912-963) sections.

A new section ("Ir anomaly and other siderophile element concentrations") was added to the supplementary information.

Additionally, some sentences have been rewritten to make them clearer and found grammatical errors have been amended.

Reviewer #1 (Remarks to the Author):

The manuscript by Rodiouchkina et al. describes an investigation of elemental sulfur and sulfur isotopes across the K-Pg boundary in marine sections from the impact crater, continental shelf and slope, and deep ocean as well as terrestrial sections from the continental US. The results are used to interpret the amount of sulfate aerosols that were formed during and shortly after the impact and their climatic and biotic impact compared to soot from wildfires and dust from pulverization of the impact terrane. The results suggest that sulfate aerosols were less voluminous than several recent studies. This is important as it provides a more accurate account of the climatic after-effects of the impact and the time frame for cessation of photosynthesis which was the likely driver of mass extinction. The authors use the results to simulate the post-impact cooling and this indicates that the initial pulse was not as dramatic as previous interpretations suggesting that the extinction took longer.

The approach is not novel in that previous studies have attempted to model both sulfur and sulfur isotope concentrations, but what the study does do is provide a more expansive dataset as well as give a potentially more holistic interpretation of the compiled data.

Overall, I believe that this manuscript is provocative and of wide interest to the community and the broader public. The issue of dinosaur extinction has substantial appeal, and this alone will fuel interest in the manuscript. However, there is one major issue that I believe needs to be resolved before the manuscript can be considered further. The key climatic events of the K-Pg impact happened so rapidly that they are often difficult to resolve in distal deep-sea sections that were deposited very slowly but offer the simplest stratigraphic interpretation. In sections that were deposited closer to the crater, key events can be more readily detected, the crucial issue is deciphering how materials were delivered to the land surface. The bulk of these thick sections were delivered by high energy processes like tsunamis and seiche waves which distort primary geochemical and climatic signals through redeposition. The key in interpreting data like the sulfur data presented here is deciphering the signals that were derived via airfall because these are the true signals that have climatic significance. And in short, I am unconvinced that the samples that are key to the interpretations made here were indeed airfall, or more precisely entirely modulated by airfall.

This appreciate this remark by reviewer 1, as this information was underexposed in the previous version of the manuscript. Indeed there are major differences in the transport and deposition mechanisms between very proximal, proximal, intermediate and distal ejecta. The

marked difference in the stratigraphy of K-Pg sites were addressed in details in Claeys et al. 2002, Schulte et al. 2010 (Fig. 2), and Smit 1999, while Alvarez et al. 1995 modeled transport to the proximal US Western Interior sites, like the ones examined in this study. Our sample selection is based on the above information and the experience gained through several field work expeditions in the Gulf of Mexico region and US Western Interior over the last 30 years. It is important to understand that the Chicxulub-generated tsunami deposition is limited to the Gulf of Mexico region, as clearly demonstrated in sites like Mimbral (Smit et al. 1992) or Brazos River (Bourgeois et al. 1989). At these sites, material can, as stated by reviewer 1, be transported by repetitive high-energy processes from the crater (see changing paleocurrent directions in Smit 1999) and disturb the S signal. This energetic sedimentation results in the following succession of K-Pg deposits: with graded clastic units (sand) on top of impact spherules (glass) at the bottom all the way to much finer material at the top (air fall), including the Ir-rich clay at the very top (Smit et al. 1992). At these sites, the Ir positive peak is often lower in concentration (< 1 ppb) than at more distal sites, spread over 30 to 50 cm and much more diffuse (with the presence of several small Ir peaks ranging from 200 pg/g to 700 pg/g, (see Schulte et al. 2010, Fig. 2) and smaller coarser layers (silt, see Smit et al. 1992 for Mimbral for example) interrupting/diluting the Ir and air-fall deposition. These tiny silty units likely result from deposition by waves generated by the last seismic movements after the Chicxulub structure has formed, and the following readjustments. It is thus likely that at these sites the deposition process affected the S signature (as for example in Brazos river), as correctly observed by reviewer 1. This is why we did **not** focus on sampling proximal K-Pg sites, except Brazos to test and verify previous data sets (Junium et al. 2022).

The intermediate continental sites, we focused on, like in the US from New Mexico to Alberta in Canada are clearly **NOT affected by this tsunami or high energy transport**; the coarse clastic sandy unit is absent; the K-Pg is only composed of the cm-thick air-fall layer (Ir with value $\gg 1$ ppb) resting on top of a layer of impact spherules. This represents the famous dual layer as Gene Shoemaker termed it in the mid 1980. Tanis (De Palma 2019, Fig. 2) is the notorious exception. Below the clearly visible air fall unit, a unique seiche deposit occurs, this is what makes the site so special (De Palma et al. 2019; During et al. 2020; Senel et al. 2023). However, this high energy unit differs **drastically** from the one recorded at proximal sites, as it does not transport all the way to Tanis (> 3000 km) material from the crater and the tsunami waves from the Gulf of Mexico likely do not reach this location. At Tanis, a “local” seiche, generated by seismic shaking triggered quickly after impact, is indeed capable of high energy deposition, well before the arrival of the air fall deposition (that covers it), which is clearly distinguished in the field. Here, the K-Pg clay (air fall) is not disturbed by

high energy sediments, and is again only a few cm thick. For details on the formation of seiche and distinction with tsunami see Bondevik et al. 2013, <https://doi.org/10.1002/grl.50639>. At Tanis, the Ir peak is clear and pronounced at concentration levels that are much higher than at proximal sites, and not diluted by silt layers from higher energy deposits (Fig. S6). At all intermediate sites, the K-Pg fine clay unit (air fall) is most likely free of reworking based on sedimentological observations, and without distortion of the primary geochemical and climatic signals through redeposition based on the recorded profiles, including Tanis. Therefore, these sites are ideal for the study of S signatures carried out in this study. The homogeneity of the obtained S values for the various characterized intermediate terrestrial sites supports this point, very well.

The authors explain the highly negative sulfur isotope signals at the shelf and deep-sea sites as a result of sulfate reduction, though the Brazos section contains a positive anomaly which is different from the other sites. The authors interpret this as a combination of a mixture of deposition from airfall and redeposition likely from tsunami and seiche waves that traveled to the shelf. At the same time the authors interpret the sulfur data from the Tanis and other western Interior sections in terms of primary airfall when most of the Tanis section at least is thought to be deposited by seiche waves as described in the initial DePalma et al (2019) publication. DePalma et al. did interpret several horizons to be primary airfall but that interpretation was not entirely convincing to me. As far as I know the other western interior sections have not been confirmed in detail to be airfall either.

We disagree with the interpretation by DePalma et al., see model of Alvarez et al. 1995., and all subsequent models of ejecta deposition. Our interpretation is supported by sedimentological (based on stratigraphy, palynology, and grain size), geochemical (Cr, Co, Ni, Ir, HSE concentrations) and isotopic (sulfur and carbon) data (cf. above).

The authors state in line 306 that terrestrial sections are likely to be only influenced by the airborne component and are therefore more reliable. However, I have not seen any proof of this here or elsewhere. I wonder whether the signals reported from Tanis and the other sections could represent mixing and dilution of the atmospheric signal by the seiche waves. This is so central to the overall interpretation that I do not think the manuscript can go forward without a more convincing interpretation of the mode of deposition in the Tanis and other terrestrial sections. Note that I have the same issue with the recent Senel et al. (2023) paper on silicate dust from the Tanis section.

It is highly unlikely, and geologically very complex to have seiche or tsunami waves affecting continental deposits at the locations from New Mexico to Alberta, especially the more northern ones, located very far away from any water bodies, where such a seiche could be generated, see paleogeographic map. In short, there is ample work and evidence for the interpretation of the K-Pg sites of the Western Interior (See Bohor et al. 1984) and Canada as the result of continental deposition, based on sedimentology, palynology, paleontology etc. (see Nichols et al. 1988; Nichols et al. 1990; Sweet & Barman, 1992, Hartman et al. 2002). No high-energy deposition has ever been reported (except for some fluvial deposits above the boundary, see famous Palaeocene dinosaurs controversy in Hell Creek Fastovsky <https://doi.org/10.2307/3514678>, Smit & Kaars, 1987 vs Rigby et al. <https://doi.org/10.2307/3514679>) at these K-Pg boundaries, which contain dinosaur-rich assemblages below and are often covered by continental coal deposits, clearly supporting undisturbed, low-energy continental deposition. This is further supported by the geochemical and isotopic records (cf. above). We have tried to make this more clear in the main text of the manuscript.

- Bohor et al. DOI: 10.1126/science.224.4651.867
- Hartman et al. <https://doi.org/10.1130/SPE361>
- Nichols, D. J. & Fleming, R. F. 1988. Plant microfossil record of the terminal Cretaceous event in the western United States and Canada. In *Global catastrophes in Earth history: an interdisciplinary conference on impacts, volcanism, and mass mortality*, pp. 130-131 (Snowbird, Utah, October 20-23, 1988; Lunar and Planetary Institute contribution No. 673).
- Nichols, D. J., Fleming, R. F. & Frederiksen, N. O. 1990. Palynological evidence of effects of the terminal Cretaceous event on terrestrial floras in western North America. In *Extinction events in earth history* (eds Kauffman, E. G. & Walliser, O. H.), *Lecture notes in Earth Sciences*, 30, 351-364 (Springer-Verlag, New York).
- Smit, J., W. A. v. d. Kaars, et al. (1987). "Stratigraphic aspects of the Cretaceous-Tertiary boundary in the Bug Creek area of eastern Montana, USA." *Mem. Soc. Geol. France* 150: 53-73
- Sweet and Braman [https://doi.org/10.1016/0195-6671\(92\)90027-N](https://doi.org/10.1016/0195-6671(92)90027-N)

If most of the sediment in the terrestrial sections was delivered by seiche waves, the question is where did it come from? Specifically, where did its sulfur come from? Was it contemporaneous sediment containing sulfate aerosol fallout from the impact (ie airfall) redeposited locally, or was the sulfur derived from other sources including older sediments or exposed bedrock, or was it a mixture of the two sources. Bottom line is that there needs to

be a discussion of the significance of the isotope signals with respect to the mode of deposition and a more convincing proof that the signals are indeed airfall.

Based on paleogeography, seiche waves are indeed difficult to generate, this is clearly NOT the case, see justification above.

The authors have also collected data from sites within the crater itself to determine the sulfur chemistry of the target. The results show great heterogeneity which is to be expected. One thing that I missed however was the interpretation of the IODP site M077. This site contains both pre and post impact rocks and I'm a little confused whether there is any significance of the post impact record. The data in Figure 1B aren't plotted at high enough resolution to determine this.

Thank you for your comment. The purpose of including a more in-depth look at the M0077A drill core in this study was that it had previously been observed to be largely devoid of anhydrites and gypsum compared to all previous drill cores. This heterogeneity is interesting to study when numerically modelling the impact as well as for us when understanding the S fingerprint of the target (for example in the case of the Yucatán 6 samples). The TRIS fraction, bulk $\delta^{34}\text{S}$, and sulfide specific $\delta^{34}\text{S}$ values for M0077A indicate the presence of anhydrites or gypsum within these cores (particularly within the graded suevite section) - likely the primary host of sulfur at this part of the impact crater - but that post impact processes cannot be ruled out due to the low S concentration observed in these sections. We therefore cannot be sure whether the obtained S signature is representing the true target bedrock, particularly as significantly lower bulk $\delta^{34}\text{S}$ were observed in the bedded suevite (- 7.4 to -5.3 ‰) sediment sections right above the graded suevite section (Table S1 in the SI). Similarly low $\delta^{34}\text{S}$ values have been observed before in the post impact sections of the M0077A drill core and have been suggested by Kring *et al.* 2020, 2021 to result from late-stage microbial reduction of S in the impact-generated hydrothermal systems. How can we then be sure that the observed $\delta^{34}\text{S}$ values for the graded suevite section are unaltered by these post-impact processes that would result in lower $\delta^{34}\text{S}$ values than for the pre-impact bedrock? For this reason, we also decided to include a small set of post-impact sediments in the record to confirm post-impact effects on the S signature recorded in this drill core. An in-depth explanation can be read in the SI, page 3. We tried to make this more clear in the main manuscript by adding two additional sentences in Lines 229-234.

Finally, I wonder if the authors should be presenting data (Figure 2B) that have little

significance for the study for example the data from Caravaca and Stevns Klint. I would prefer to see the data from the other terrestrial sections even if they are from other sources.

We agree with the reviewer and have added the S profiles of all previously published terrestrial sites (Dogie Creek, Brownie Butte, Knudsen Coulee, and Knudsen Farm) to Fig. 2E-I. However, we prefer to keep the S profiles of the marine sites investigated in this study as this provides a holistic presentation of the available S isotope records across the K/Pg boundary. In our view, it is also important to include marine K-Pg sites, like Caravaca, Stevns Klint, and Brazos River to show that the S record is disturbed at these locations and that these records cannot be used for S mass balance considerations. Brazos River is of particular importance as the site has been used previously for impact released S estimation (Junium et al. 2022) and a discussion of the applicability of this site and other marine sites in this context is important.

Minor points

Line 68 also carbon from buried petroleum (Kaiho et al. and Lyons et al. 2020)

Thank you. We have added this information.

Line 87-90 need references for the models.

We have added 3 references for paleoclimate model studies focusing on the influence of the impact-released S.

Line 265 don't you mean a positive isotope anomaly?

Yes, thank you. We have added the word "anomaly" instead of "value".

Line 273, repetitive with sites named, not necessary

While we agree with the reviewer that this information is superfluous, we are of the opinion that this repetition makes it more clear to the reader what sites are being discussed.

Otherwise readers would have to scroll up to Line 256 and 259 (from Line 320) to understand which sites we are referring to.

Sulfur isotopes and concs in Figure 2 needs Ir data for airfall

Thank you for the suggestion. While we agree this would strengthen our case, we have decided that Figure 2 would become too overloaded if we include both the profiles of previously published sites (Dogie Creek, Brownie Butte, Knudsen Coulee, and Knudsen Farm) as suggested and the siderophile data suggested by you. Instead we have opted to a section in the SI called “**Ir anomaly and other siderophile element concentrations**” and we have added the requested information in the manuscript to connect the S anomaly to the siderophile elements and previously published median grain-size data (Senel et al 2023) in Lines 262-263, 278-282, 297, and 311.

Figure S2 should have labels on to indicate which section is which

Yes, we agree, this makes it more clear. This information has been added to Fig. S2.

Reviewer #3 (Remarks to the Author):

Review of "Reduced contribution of sulfur on the mass extinction associated with the Chicxulub impact event" by Rodiouchkina et al.

This paper uses measurements of sulfur (S) concentrations and isotopic compositions from terrestrial sections across the K-Pg boundary to estimate the emissions of S-bearing gases from the Chicxulub impact associated with the end-Cretaceous mass extinction. Since the amount of S-bearing gases from the impact is difficult to quantify from geophysical simulations of the impact process, but highly relevant for the climatic effect of the impact, this paper adds an important piece of information required for a better understanding of the end-Cretaceous extinction event and definitely warrants publication.

That being said, there are severe issues with the climate modelling part of the paper which imply that the manuscript cannot be published in its present form and requires major revision.

Major comments:

(3.1) In several places, the model is described as a "General Circulation Model (GCM)" (e.g. in lines 162-163, 753). By today's standards, this would imply an AOGCM representing the full three-dimensional dynamics of both the atmosphere and the oceans. In this case, however, the model is "just" an atmospheric general circulation model (AGCM) coupled to a strongly simplified ocean model which will affect the results (see below). This should be mentioned and discussed in the main text of the paper, not just in the methods section. Also, I would not call this model "state-of-the-art" (line 162) since it is not an AOGCM.

We agree with the point of reviewer that the 'state-of-the-art' would better be represented by AOGCMs with coupled 3D ocean-atmosphere models. Thus, we dropped the 'state-of-the-art' phrase, as we combined a 3D atmospheric GCM (based on planetWRF general-purpose planetary atmosphere model; Richardson et al., 2007) with a simplified ocean mixed layer (OML) parameterization (Pollard et al., 1973, Davis et al., 2008, Nellipudi et al., 2021). We added model description, validation, and limitations in Lines 885-887 and 912-963.

(3.2) Even in the methods section, there is a certain lack of details regarding the model.

We added more details about the model description in **Section Methods: Description of the paleoclimate modeling simulation**. Regarding ocean modelling, please refer to Lines 885-887 and 912-963.

How thick is the mixed layer in this model?

The initial condition for the mixed layer depth ($h_{\text{omi}}=100$ m) and the deep layer thermal lapse rate ($\Gamma_{\text{omi}}=0.14$ K/m) are added to the Methods (Lines 929-930).

How is meridional heat transport in the ocean prescribed since it obviously cannot be simulated dynamically?

Added in Lines 917-925.

What about the sea ice model, does it capture the dynamics of sea ice or just the thermodynamics? Again, this is important to be able to judge the reliability of the climate modelling results presented in the paper.

Following the comment, we added a description in Lines 931-963.

I think something might be wrong with the "global surface temperature" maps presented in Fig. 5. I am particularly worried about the relatively high temperatures in the polar oceans after the impact. I suspect that the temperatures could be surface rather than surface *air* temperatures which would explain why the polar oceans (which should be covered by sea ice) are only at temperatures at or above the freezing point of seawater rather than at temperatures similar to continental areas (which is what I would expect for cold air above sea ice in polar regions).

We presented "surface temperatures, T_s " on land and ocean across both the main text and supplementary, not the "surface air temperatures" (T_2 variable output of planetWRF, referring to air temperature at a 2-meter elevation above continents and oceans). As pointed out by the reviewer, surface temperatures (T_s) explain why polar oceans are closer to the freezing point of seawater.

If the temperatures shown in the maps are indeed surface *air* temperatures, something could be wrong with the treatment of sea ice in the model. In any case this needs to be double-checked and/or discussed, in particular with respect to the emphasis on global mean temperatures being above the freezing point throughout the paper (e.g. lines 55-56, 370-372, 388-389, Fig. 6).

As mentioned above, they are rather "surface temperatures, T_s " on land and ocean. We clarified this throughout the main text and supplementary information files as suggested.

Please note that other model studies present surface *air* temperatures, of course, so this is relevant for the comparison with other studies and proxies - and for the sweeping interpretations in the paper regarding species survival.

Regarding global temperature analysis, we examined the difference between surface temperature (T_s) and surface air temperature (T_a) for the year preceding the impact. While the global average surface temperatures consistently displayed a 1-degree difference (Figure S17a), the variances can be larger regionally (Figure S17b). Here, we opted to keep T_s in the manuscript as it serves as a better diagnostic for proxies and surface environments, offering a better representation of ground conditions vital for sustaining life after impact including plants, seeds, and burrowing organisms. It also facilitates direct comparisons with our recent paper (Senel et al., 2023) investigating various ejecta scenarios. Moreover, we expect minimal difference between T_s and T_a post-impact, given the extremely cold conditions and stable boundary layer formations across continents. In the context of the present winter evolution of stable boundary layers (SBL) over Antarctica, for example the 10 years of temperature observations from a 42m tower in the East Antarctic plateau (Ghenton et al., 2021) indicate minor temperature changes ($\sim 0.2^\circ\text{C}$ in summer and $\sim 1^\circ\text{C}$ in winter) within the first 2 meters of the stable boundary layer, corresponding to the atmospheric surface layer. (Note that strong thermal inversions can develop at the upper layers (Ghenton et al., 2021) especially in calm winter conditions, resulting in $>20^\circ\text{C}$ warmer temperatures at the top of the tower (free-atmosphere). As an example, while the ground temperature was -78°C , the air temperature became -77°C and -55°C at 2 and 42 meters, respectively).

By the way, the continental areas appear to be at temperatures below freezing even for this model study, so I fail to see how the results presented in this paper would lead to significantly different conclusions regarding species survival compared to other studies.

Please see Lines 460-475. Please also note that for a species to go extinct, nearly all individuals of this species have to die. If a critical number of individuals persists, enough to form a viable population, a species survives. Undoubtedly, the severe cooling in the continental interiors will have caused many animals and plants to die. Yet, our results suggest that the lower mass of S released meant that even at the peak of the impact winter, temperatures remained above freezing in many coastal regions (see Fig. 5), providing refugia for surviving terrestrial species.

Finally, are there really regions with annual mean surface (air?) temperatures beyond 40 degrees for a 560 ppm pre-impact state? This would be really surprising - and add to my

suspicion that the limitations of the model might be more significant than the authors care to admit... These extremely high sea surface temperatures in the tropics could be due to a lack of meridional heat transport maybe?

Not exceeding 40°C for the 560 ppm pre-impact state, please see our pre-impact comparison in Figure S16. The annual mean surface temperature is $T_s \lesssim 38^\circ\text{C}$ (Figure S16B), while the surface air temperature (at 2 meters) is $T_a \lesssim 34^\circ\text{C}$ over tropical oceans (Figure S16). It is worth noting that regional surface air temperatures (T_a) are 4-8°C cooler than the ocean surface (T_s) in the tropics (Figure S17b). This is most likely because T_a absorbs the atmospheric response being more susceptible to changes induced by atmospheric boundary layer (ABL) processes (e.g., wind shear, turbulence, buoyancy). Additionally, surface air temperatures (at 2 meters) are up to 5°C warmer than the ocean surface (T_s) in polar oceans.

(3.4) The lack of a full ocean model (and I suspect also of a dynamics sea-ice model) will certainly affect the results. For example, in this model study the lowest temperatures are reached relatively early, and the recovery is faster compared to studies using full ocean models (e.g. lines 360-361, 364-365). This is most likely due to a lack of the full thermal inertia of the ocean in model. The influence of model limitations on the results definitely needs to be discussed.

Thanks for this suggestion. We included a discussion on the limitations of our model in the **Section Methods: Description of the paleoclimate modeling simulation**. Regarding the earlier onset of initial cold and relatively fast recovery, please refer to Lines 485-492.

Minor comments:

Fig. 5: I would also suggest to use a colour scale in which the freezing point is white rather than blueish to make it easier to see which regions are below freezing in the figure.

Done.

The discussion in lines 397-409 focusses solely on Senel et al. (2023) but should include other relevant studies, in particular Tabor et al. (2020), but also Brugger et al. (2017).

We included a discussion addressing those studies in Lines 480-492.

Reviewer #4 (Remarks to the Author):

In this study, Rodiouchkina and colleagues report S-isotopic data and sulfur contents from various terrestrial and terrestrial-marine sections across the K-Pg boundary. The authors developed a simple mass balance model to estimate the amount of impact-vaporized sulfur. The S-isotopes and sulfur contents are two major variables in their model calculations. The authors argue that their estimate of vaporized sulfur from the Chicxulub impact is approximately five times lower than that proposed by Artemieva et al. (2017), but it aligns with numerical estimates from the 1990s. Subsequently, the authors utilized the same GCM model they published last year (Senel et al., 2023; ref. 12) to discuss the cooling effect caused by sulfate aerosols and their implications for the demise of the non-avian dinosaurs. The geochemical measurements are in high precision and the paper is well written. The results presented in this study contribute a new piece of evidence for our understanding of how dramatic climate change can lead to mass extinction in the geological past and potentially in the future. I have a few major comments which may be helpful when the authors revise their manuscript:

1: S-isotope analysis constitutes a significant component of your presentation. However, S-isotopes are nearly absent from your introduction (lines 59-163). I encourage the authors to enlighten readers on how S-isotopes can assist in estimating vaporized sulfur from the Chicxulub impact. Highlighting the novelty of your empirical estimate could also provide valuable context for the readers. If you find the introduction too lengthy, consider condensing the summary of previous work (lines 91-117) for brevity.

Thank you for the suggestion. We agree that additional background information on the use of S isotope ratios in this context are needed in the introduction. Following the suggestion we have therefore added section between 153-159 and Lines 163-171. We decided not to condense the summary of previous work, as we are of the opinion that it is important to summarize what has been done before in terms of estimating the amount of impact released S as well as climate models related to these estimations. For the reader, it is critical to have this information to fully understand the importance of this study, the novelty of this study, and to be able to position Fig. 3 in its context.

2: I agree with the authors that a positive peak in sulfur concentration, along with a positive $d^{34}\text{S}$ value, may be a primary signal resulting from the impact. I am also convinced that the negative $d^{34}\text{S}$, coupled with an increase in sulfur contents from Stevns Klint, Caravaca, and Brazos River, does not exclusively result from atmospheric sulfur deposition. It is correct to

suggest that 'This is likely a result of S fractionation during microbial reduction of sulfate followed by pyrite sedimentation' (lines 260-261). However, it is important to note that sulfate deposited from the atmosphere in rivers and lakes can also undergo biological reduction to form pyrite.

We appreciate this suggestion and agree that the S deposited from the atmosphere in the terrestrial sites could also have been affected by biological reduction and pyrite formation. However, based on the S isotope ratios observed and the estimates obtained for the different sites we consider this effect to be relatively small for the terrestrial sites included in this study. The estimates of the total amount of impact-released S obtained for the terrestrial sites of Tanis, Knudsen Coulee, Knudsen Farm, Dogie Creek, and Brownie Butte are highly reproducible (67 +/- 39 Gton at the 2SD level or 20 Gton at the 1SD level). If biological reduction was such an important factor in steering the d34S values, this would undoubtedly have led to strong local lacustrine phenomena altering the recorded atmospheric signal, possibly associated with siderophile element fractionation (e.g., Goderis et al., 2021), which we don't observe. A sentence Lines 390-393 was added to address this.

I suggest the authors to provide a more detailed explanation of their arguments regarding the origin of the negative d34S shift. Several factors can influence the S-isotopic compositions of biogenic pyrite, and it seems that an increase in sulfate concentration from seawater input is a plausible candidate, if not the only one (see a recent study by Xu et al., 2023, Chemical Geology 642, 121790). Readers would benefit from a more comprehensive interpretation in addition to citing previous work.

We thank the reviewer for this suggestion. The origin of negative d34S shifts at the marine sites are expanded upon in Lines 322-326. The mechanism proposed by the reviewer could indeed be a plausible explanation for the S anomaly observed for the terrestrial sites if we indeed saw large negative d34S shifts for any of the terrestrial sites as observed in the recent study by Xu et al., 2023 (shift up to 50 per mil). As mentioned above we also see very similar S profiles for all of these regardless where the site is situated in relation to seawater as well as different local environments. We are grateful that you lifted this potential source of error that was not sufficiently discussed within the text and new paragraph between Lines 331-354 regarding it has been added to address it.

3: I suggest that the authors elaborate on their model construction in the section from lines 298-306. While it appears concise, this aspect is a crucial contribution to your study and could benefit from a more detailed explanation.

Thank you for the suggestion. We have expanded on the method in the introduction in Lines 153-159. However, following the journal's format requirements we cannot explain the methodology in more detail in the Introduction or Results and Discussion section as a more detailed explanation is included in the Method section of the manuscript.

4: As I'm not a GCM model person, I am unable to provide comments on the validity of the model and the simulated results. I presume that other reviewers with expertise in this area can offer valuable insights.

Yes, we have received constructive suggestions from another reviewer and have corrected where deemed necessary.

Response to the reviewers - Nature Communication manuscript NCOMMS-23-53601:

“Reduced contribution of sulfur to the mass extinction associated with the Chicxulub impact event”

Katerina Rodiouchkina, Steven Goderis, Cem Berk Senel, Pim Kaskes, Özgür Karatekin, Michael Ernst Böttcher, Ilia Rodushkin, Johan Vellekoop, Philippe Claeys, Frank Vanhaecke

Dear reviewers,

First of all, we would like to thank you again for your time and effort reading through this manuscript and for your detailed comments and remarks. Please find the thoroughly reworked PDF document with all changes based on the given comments and suggestions marked through track changes. Below, a summary of the implemented changes is provided followed by a point-by-point response to each comment, as requested. We hope that our responses fully address the comments received during the reviewing process. The response from the authors is highlighted in the color blue.

Yours sincerely,

Katerina Rodiouchkina, on behalf of all authors

Summary of implemented changes:

Introduction:

- Discussion about previous numerical estimates has been expanded to provide a more clear background of the reliability of the different estimates, Lines 91-148.
- More in-depth explanation about the importance of accurate shock pressures for numerical simulations has been added in Lines 165-178.

Result and discussion:

Sulfur concentration and isotopic composition in the target:

- Rewrote some sentences to make the text more clear

Sulfur concentration and isotopic profiles for K-Pg boundary sites:

- Added Lines 301-305 to have a more seamless connection to the previous section of the manuscript according to the comments of Reviewer #1.
- Moved the siderophile element anomalies discussion from the supplementary information to this section of the manuscript following the recommendation from Reviewer #1. Figure 3 was added to the manuscript from the supplement.

Estimated total amount of impact-released sulfur:

- Expanded on the discussion of the comparison between the empirical estimate obtained in the current study with previously published numerical estimations, Lines 501-547.
- Removed one estimate from Figure 4 as we gained new information that this estimate was not novel and based on estimates already included in Figure 4.

General:

- Removed all sections and Figures related to the climate modelling results from the previous version in both the manuscript and the supplement.
- Additionally, some sentences have been rewritten to make them clearer and found grammatical errors have been amended.

Reviewer #1 (Remarks to the Author):

This is the second time I have reviewed this manuscript and I'm incorporating the author's detailed comments on my first review in my analysis. Overall, I believe the authors have done a very good job of addressing my previous concerns. This is a very extensive piece of work both in the number of analysis, and the depth and extent of the interpretation, so it's really hard to go justice to all of the science in a short format manuscript. I think that is what gave rise to my previous concerns and misunderstandings and that is what might still drive some of them.

In summary, in my mind this is a very impressive and important piece of science that deserves to be published. I'm more convinced the interpretation of S volume is realistic than I was before. I do have some doubts about the Tanis section as I describe below, and I think the authors could continue to resolve some of them, but it seems that there is somewhat less focus on Tanis here than in previous version which may just be a perception. Overall, I support publication of this provocative paper with some minor revisions. As I said before I think that this novel interpretation of the role of sulfur gasses will drive discussion on extinction mechanisms and recovery in a positive direction.

We thank the reviewer for this fair and constructive evaluation.

The authors have presented so much data and again I congratulate them on this. They cover a lot of ground in a relatively short manuscript including the detailed sulfur isotope geochemistry and climate modeling based on it that are central to their main argument. Their data and interpretation therefore spill over into a lengthy supplement. This concern is not unique to this manuscript, its true of many papers that end up in short format journals. Being honest I just skimmed the supplement for information om Tanis and this brings me back to one of my other concerns in the first review. There are many possible ways of telling this story admittedly and I hope my suggestions below will make it clearer.

We agree with the reviewer, and the required substantial restructuring of the manuscript in this round should hopefully have tackled some of these concerns.

The authors have responded to my concerns that the Tanis section includes redeposition

from seiche deposits in addition to airfall deposits. They rely on the Ir and other platinum metal concentrations to make the case for airfall in the absence of detailed sedimentology (I'm assuming the palynology they refer to here is published).

The palynology of the Tanis tonstein, albeit briefly, has been described in the Supplementary Material of DePalma et al. (2019).

"A thin ~6 cm carbonaceous bed overlying the KPg tonstein (**Figs. 3, S1**) revealed an impoverished palynofacies dominated by ferns (>67%), comprised mostly of *Cyathidites* and *Laevigatosporites*, with minor *Azolla* and *Deltoidospora*. This correlates with the well-documented "fern-spike" that occurs in strata immediately succeeding the impact layer (**16**). The succession of the KPg tonstein; an organic-rich lignitic stratum preserving the fern-spike; and an overlying basal Paleogene sandstone body is a classic terrestrial manifestation of the KPg boundary as preserved throughout the Western Interior and directly correlates with the stratigraphy of nearby KPg sections such as Mud Buttes. "

In addition, the authors have added some text (Lines 331-354) on the Western Interior palludal and lacustrine terrestrial sections in general and the possibility of sulfate reduction in them. I really appreciate this but I suggest they give more detail on the background of what is driving the interpretations of the sedimentology of these sections as a group. Much of the audience like myself are unfamiliar with this literature and I think it would help the manuscript enormously.

The deposition of the K-Pg sequence in the Western Interior has been known and described in detail since the mid 1980s in more than 20 papers (e.g. Bohor et al. 1987, *Geology*, 15, 896; Nichols & Fleming, 1988, *LPI* 673, 130-131; Nichols et al. 1986, *Palynology*, 11, 246; Pilmore et al. *Science*, 223, 1180). We are of the opinion that more detailed interpretations fall outside the scope of the current manuscript, but we have added some sentences in lines 435-441 to make these aspects more clear.

This is relevant because presumably the coal unit must have been deposited over many years and not just from airfall (lines 281-283).

The reviewer's main concern is that the coal could be much older than just a couple of years, possibly based on compression of peat layers that could take considerably more time. However, we rely on both new and published PGE data, in conjunction with published palynological data that in our opinion both suggest deposition of the coal layer (and hence the sulfur isotope anomaly) in just a couple of years to tens of years maximum. We attach a published figure from Schulte et al. (2010) that shows the iridium anomaly and (most of the) fern spike in the c. 4-5 cm thick coal layer at Starkville as an example.

Figure S17: The K-Pg boundary in the Western Interior. (A) Schematic lithology of the K-Pg boundary interval at the Starkville North site, 5 km south of Trinidad, Colorado. The large black dots and the solid line show the iridium concentration, the open circles and the dashed line show the fern spore percentages (modified after S52). (B) Polished surface of the dual-layer K-Pg boundary claystone at the Teapot Dome site, Wyoming. Green arrows show ejecta spherules that are now altered to alumino-phosphate (gorceixite, goyzite) in the lower claystone. Black arrows show detrital quartz (shocked and unshocked) that is abundant in the upper claystone. Two large rip-up claystone clasts are marked with "R". Photo courtesy of Glen A. Izett. (C) Polished surface of the K-Pg boundary interval at the Clear Creek North site, Colorado. The lower K-Pg claystone bed consists of kaolinite clay and minor amounts of illite-smectite (I-S) mixed-layer phyllosilicates. Typically, this bed contains shreds and deformed laminae of vitrinite. This bed also contains ejecta spherules altered to kaolinite and alumino-phosphate. The upper part of the K-Pg claystone couplet is again compositionally and texturally different from the lower bed. It consists predominantly of illite-smectite (I-S) mixed-layer phyllosilicates and minor kaolinite and contains an assemblage of detrital silicate mineral grains (shocked and unshocked). Highest concentrations of the platinum group elements (PGEs) are usually present in the upper layer. Photo courtesy of Glen A. Izett.

The sedimentology of Tanis itself has not been investigated in detail and given the importance of the interpretations that are being made from this section here and elsewhere a detailed investigation of process sedimentology really needs to happen. I'm not saying this should be in the scope of the current study though. The authors themselves have not been

able to document the sedimentology in further detail because they could not analyze grain size in the key coal interval (Figure S6). Down the road there needs to be more detailed sedimentology.

We certainly agree with the reviewer that more detailed sedimentological studies required for the Tanis site in the near future. We have added an addition to the sentence on Line 348 “although further sedimentological studies and grain-size measurements in the coal interval should be carried out.”. However, we are of the opinion that the Tanis site still yields a reliable estimate of impact released S. Our argument of airfall is not solely based on grain size, but also the S anomaly and siderophile element enrichment consistent with derivation from the target rock and impactor. We are also confident in the estimate of the amount of impact-released S from this site, as it presents (i) a realistic value that (ii) matches up with estimates from 4 other sites located in USA and Canada.

Finally given the importance of Tanis to the overall interpretation, I think it would be more convincing to have a stand-alone figure in the main text on this section, in particular, including the Ir and other REE data in the main text not the supplement.

We agree with the reviewer that the Tanis site is important for the overall interpretation and for the final empirical estimate but consider the other 4 terrestrial sites equally important for the final estimate. The discussion on the marine sites is also important. To accommodate the reviewer and following the major overhaul of the manuscript, we have therefore moved the siderophile element discussion from the supplementary information to the main text and made Figure S6 for the Tanis site Fig. 3 in the main manuscript.

I've spent some time looking at the Junium et al. paper that focused on Brazos and that was one of my previous concerns. I now agree with the authors' interpretation of this dataset, the key sulfur isotopes interpreted by Junium were from the tsunami interval and thus likely a combination of sources, (largely) transported in from the crater as well as (minor) from airfall. So I agree with the interpretation that Junium et al. have overestimated the volume of sulfate aerosols. I do think that there needs to be more discussion about the deposition of the post tsunami upper units at Brazos (E, F and G) in Figure and how the S values differ from the tsunami beds. In addition, I think that the authors need to make it clearer how their Brazos analyses differ from those of Junium et al.

Thank you. From the Junium et al. paper, it was unclear how the estimate of amount of impact-related S was obtained. We had contacted them prior to the first submission to understand the methodology, but we never received an answer. After emailing more of the authors of the Junium et al. paper, we received an answer during this round of revisions. The presented estimate is not an empirical one based on their S profiles for the Brazos River. We had misunderstood this in the initial submission. Rather, their estimate is a unit conversion based on previously published numerical estimates of 30 to 540 Gt S (Pierazzo et al, 2003). Considering this information, any references and discussions of the Junium et al. estimate has been removed from the current version of this manuscript, as it is not novel nor original and based on prior work that is already cited in the current version of the manuscript.

Minor suggestions

Line 121, "risking possible overestimation" is unclear. Does this mean the authors didn't consider these factors or the fact that they didn't risked overestimation?

The authors didn't consider these factors, which risks overestimation. However, this sentence was removed and is discussed further down in the manuscript more thoroughly.

Line 151 Do you mean the impact crater itself?

Yes, we meant the impact event. This has been added.

Lines 171-174 The statement about Brazos being a tsunami deposit depends on where Junium et al.'s measurements are made since some of the record is fallout from suspension and storm deposits. So they need to clarify that Junium et al.'s measurements are from the tsunami interval at Brazos.

See comment above about the estimate provided by Junium et al.

Line 252 Should have a connection with previous section for example say that comparing global boundary sites with the target sulfur.

Two sentences have been added to connect this section to the previous section.

Lines 253-263 Should give figure numbers for the individual sections (ie. Fig 2B-2E)

Figure numbers have been added for the individual sites.

Line 262 Weren't some of these measured in previous pubs?

Yes, we have moved the siderophile element discussion to the main manuscript and referred to previous papers in which Ir anomalies have been measured for some of the sites.

Line 259 "sampled in the same region as the one previously used" this is unclear.

This is removed as the Junium et al. estimate is no longer relevant for the current manuscript.

Line 280 Don't you mean Figure S6 for grain size?

Yes, we meant to write S6. Figure S6 has been moved to the main manuscript in this version and is now called Figure 3.

Lines 322-326 Recommend this sentence be divided in two.

Thank you, this sentence has been rewritten to make it more clear.

Line 329 "cannot be ruled out" is a little weaker than previous text in the paragraph which implies sulfate reduction fairly strongly.

We have changed it to highly probable instead.

Line 337 Addition of sulfate from where? Seawater incursion?

Yes, we have included seawater incursions as an example in the sentence.

Line 342 "the large $\delta^{34}\text{S}$ shift" what does this refer to? The K-Pg or other impacts?

The large shifts discussed in this section were indeed related to other impact craters. After internal discussion, we have realized that the sentence "Sulfur isotope fractionation between sulfate and sulfide, and gypsum and marcasite between 44 and 78‰ was observed at

Haughton Crater in the Canadian High Arctic, and a sulfur isotope spread of 77‰ was reported for the Miocene German Nördlinger Ries crater” might be confusing within the terrestrial site discussion as these papers are focusing on large S isotope shifts observed at other impact sites. This type of large S isotope shift has also been observed for the Chicxulub impact site, so this strengthens our observations (Schaefer *et al.* 2020, and Kring *et al.* 2020, 2021). However, this sentence does not fit in its current position and has therefore been removed and added to the supplementary information instead.

Line 374-376 Also these sites are condensed so only give very long-term estimates, terrestrial sites are more expanded

Yes, a sentence at line 481 has been added to address this.

Line 399 Be specific what the post depositional effects are.

Thank you. A clarification was added.

Line 460-477 This is more important. This paragraph should be reversed with following paragraph that specifies that soot and silicate dust are smaller drivers than sulfur. Without this context the impact of sulfur on its own is confusing. There needs to be one stand alone paragraph focusing on the implications on life at the end of the manuscript.

Unfortunately, the section regarding the climate modelling performed in this study has been removed from the manuscript per suggestion by another reviewer and the editor. We will continue to work on these aspects in the upcoming months. A large portion of the lines you have indicated remain and now compose the final paragraph of the manuscript.

Figures

Figure 2 need to plot S concentrations in all of the panels at the same scale to make the changes more comparable. I would suggest this for the isotopes too because it would be more graphic to have the positive and negative shifts going in different directions even though it may be hard. Also, the red dashed line is the base of the K-Pg boundary section.

We have tried to implement these changes, but this leads to a loss of visual information, as the dimensions and axes scales are distinct for each of the sites. As an example, we have plotted the S concentrations using the same scale in the first figure below and as you can observe differences within a K-Pg deposition site are hard to see, especially for Caravaca. We also plotted the S delta values using the same scale in the second figure below and

using the y-axis as the zero point. The differences within one site are hard to see in many of the K-Pg sites. The combined figures will become very messy if we put the y-axis at the zero point for the delta values. These graphs are first and foremost there so we can compare differences in the S concentration and isotope ratios over the K-Pg boundary sedimentary section. This is hard to do if the same scale is used everywhere. The values remain in the SI. We therefore think our data is best represented in the fashion that they are right now in the current Figure 2.

S concentrations plotted using the same scale.

S delta values plotted using the same scale and using the zero point as the axis.

Reviewer #3 (Remarks to the Author):

Review of the revised version of "Reduced contribution of sulfur to the mass extinction associated with the Chicxulub impact event" by Rodiouchkina et al.

The authors have now taken into account most of my comments on the climate-modelling part of the paper. However, the additional details in the model description only confirm my suspicion that the climate-model setup used for this study is simply unsuitable for investigating the climate of the late Cretaceous and, more importantly, the impact scenarios. I will detail my major concerns in the comments below. As it stands, the manuscript cannot be published with the modelling part included, in my opinion, or the climate modelling has to be repeated with a vastly improved model setup. Otherwise no-one in the climate modelling community would believe these results.

Also, I see from the other reviews that there is quite a bit of a discussion concerning the empirical part of the manuscript on which I cannot provide any expertise.

We acknowledge the reviewer for their analysis of our manuscript. We do not agree with this take on the climate modeling parts, as the Chicxulub impact event had extremely profound, short-term effects on the Earth system, and any simulations of this event to date remain inadequate, with meridional heat transport and sea-ice models obviously constituting important parameters but not having a dominant influence on the modeling outcome.

However, as the manuscript is unlikely to be accepted if the climate modelling part and we cannot implement meridional heat transport in the ocean and (at least) a thermodynamic sea-ice model in a timeframe of 3 months, we have decided to revisit these aspects in a future manuscript. Following a consultation with an external expert who has extensively worked on this topic, we have after careful consideration decided to remove the climate modelling part from the manuscript and focused on the empirical S estimate.

Major comments:

The climate model is not suitable for the current investigation. This statement needs some background information: State-of-the-art coupled climate models (AOGCMs) combine a 3D atmosphere model (AGCM) with a 3D ocean model (OGCM) including a sea-ice model

which captures both sea-ice dynamics and thermodynamics. As the authors point out correctly, these models are computationally expensive, so faster, simplified models are frequently used in paleoclimate studies. This is perfectly alright as long as the simplifications do not severely affect processes relevant for the study, and as long as the limitations are taken into account when discussing the results. Neither is the case for this manuscript, unfortunately.

A typical model used for paleoclimate modelling could be an AGCM (as in the case of this study) coupled to a simplified ocean and sea-ice model, for example a mixed-layer ocean model with prescribed meridional heat transport and a thermodynamic sea-ice model. This is not ideal, of course, because the meridional heat transport in the ocean cannot adjust to changes in climate in this case, and because sea-ice dynamics is important (in addition to thermodynamics), but it would be acceptable for this study.

No longer relevant.

The problem with the model setup used here is that it is not just simplified with respect to state-of-the-art climate models, but even overly simplified compared to simplified setups typically used in paleoclimate studies. There are two substantial shortcomings of the model setup used in the manuscript:

(1) There is no meridional heat transport in the ocean at all which is most likely the reason behind the unrealistically high tropical sea-surface temperatures (see also my comment on the original version of the manuscript) rather than the unconvincing explanation in the rebuttal. This will yield a physically unrealistic pre-impact climate state and a completely unrealistic response of the ocean (and the climate) in the impact scenario, making the simulations in essence meaningless.

(2) There is no sea-ice model at all, not even a simple thermodynamic model. This is a crippling limitation. First of all, I do not believe the authors' argument that there won't be any (even seasonal) sea-ice at 560 ppm and for a solar constant which is lower than today. Of course there will be sea ice (at least in winter) even in the warmer pre-impact state! Much worse, the drastic global cooling after the impact cannot be meaningfully simulated without taking sea-ice formation into account. Unfortunately, this makes the entire modelling effort rather pointless from a climate physics point of view. This is also the reason for the very warm polar regions (see also my comment on the original version of the manuscript) rather than the argument given in the rebuttal. By the way, the absence of a sea-ice model also

explains why global surface air and surface temperatures are so similar. They would not be in reality since the air above sea ice gets really, really cold...

Assuming that the empirical part of the manuscript is sound, there are two ways to move forward. Either the authors remove the climate-modelling part from the manuscript since it is unsuitable. Or the authors redo all the simulations with a model setup which includes (prescribed) meridional heat transport in the ocean and (at least) a thermodynamic sea-ice model.

We have opted for the first way forward.

Note that this is not nitpicking. The limitations of the model setup used in the study are really fundamental and not acceptable within the paleoclimate modelling community.

The other thing which worries me is that this is not the first study by the authors on the Chicxulub impact with this model setup. In my opinion, the model limitations are so severe that this reaches the point where one could think of submitting clarifying corrigenda/addenda for the earlier publications...

Additional minor comments:

l. 474: I appreciate that the authors have replaced "global temperature" by "global surface temperature" throughout but I think that it should be made even clearer that these are not surface air temperatures, e.g. by inserting "(not surface air)" or similar when temperatures are first mentioned.

No longer relevant.

l. 484-486, "No value within this new empirical estimated range of impact-released S gives rise to global-average temperatures below freezing point (Fig. 6.):": I am not sure why this is emphasised. Also, this is so severely affected by the model limitations that the statement is not supported by the simulations presented in the manuscript.

No longer relevant.

Reviewer #4 (Remarks to the Author):

I have carefully evaluated the revised manuscript. The authors have addressed my comments satisfactorily. Therefore, I recommend it for publication.

Thank you.